
# Influences of land use changes on the dynamics of water quantity and quality in the German lowland catchment of the Stör

Chaogui Lei, Paul D. Wagner, Nicola Fohrer

Department of Hydrology and Water Resources Management, Institute for Natural Resource Conservation, Kiel University, Olshausenstr. 75, 24118 Kiel, Germany

*Correspondence to*: Chaogui Lei (cglei@hydrology.uni-kiel.de)

**Abstract**. Understanding the impacts of land use changes (LUCC) on the dynamics of water quantity and quality is necessary to identify suitable mitigation measures that are needed for sustainable watershed management. Lowland catchments are characterized by a strong interaction of streamflow and near-surface groundwater that intensifies the risk of nutrient pollution. This study aims to reveal the relationship between long-term land use change and the water and nutrient balance in a typical lowland catchment in northern Germany. A hydrologic model (Soil and Water Assessment Tool, SWAT) and partial least squares regression (PLSR) were used to quantify the impacts of different land use types on the variations in actual evapotranspiration (ET), surface runoff (SQ), base flow (BF), and water yield (WYLD) as well as on sediment yield (SED), total phosphorus (TP) and total nitrogen (TN) loads. To this end, the model was calibrated and validated with daily streamflow data (30 years) as well as sediment and nutrient data from two water quality measurement campaigns (3 years in total). Three model runs over thirty years were performed using land use maps of 1987, 2010, and 2019, respectively. Land use changes between those years were used to explain the modelled changes in water quantity and quality on the subbasin scale applying PLSR. SWAT achieved a very good performance for daily streamflow values (calibration: NSE=0.79, KGE=0.88, PBIAS=0.3%; validation: NSE=0.79, KGE=0.87, PBIAS=7.2%), a satisfactory to very good performance for daily TN (calibration: NSE=0.64, KGE=0.71, PBIAS= -11.5%; validation: NSE=0.86, KGE=0.91, PBIAS=5%), a satisfactory performance for daily sediment load (NSE=0.54-0.65, KGE=0.58-0.59, PBIAS= -22.2%-12%), and an acceptable performance for daily TP (calibration: NSE=0.56, KGE=0.65, PBIAS= -4.7%; validation: NSE= 0.29, KGE= 0.22, PBIAS= -46.2%) in the Stör Catchment. The variations in ET, SQ, BF, WYLD, SED, TP, and TN could be explained to an extent of 61%-88% by changes in the area, shape, dominance, and aggregation of individual land use types. They were largely correlated with the major LUCC in the study area i.e. a decrease of arable land, and a respective increase of pasture and settlement. The change in the areal percentage of arable land positively affected the dynamics of SED, TP, TN and negatively affected BF, indicated by a Variable Influence on Projection (VIP) > 1.16 and large absolute regression coefficients (RCs: 0.6-0.88 for SED, TP, TN; -1.65 for BF). The change in pasture area was negatively affecting SED, TP, and TN (RCs: -0.69 - -0.12, VIPs >1) while positively affecting ET (RC: 0.09, VIP: 0.92). The change in settlement percentage had a VIP of up to 1.17 for SQ and positively and significantly influenced it (RC: 1.16, p-value < 0.001). PLSR helped to identify the key contributions from individual land use changes on water quantity and quality dynamics. These provide a quantitative basis for targeting most influential land use changes to mitigate impacts on water quality in the future.

## 1 Introduction

Good water quality and quantity are essential for enhancing ecological stability and diversity, and both of which play important roles in maintaining sustainable agricultural or economic development and human health (Antolini et al., 2020; Gleick, 2000; Lu et al., 2015; Singh et al., 2017; Srinivasan and Reddy, 2009). The dynamics of water quality and quantity at the catchment scale are mainly governed by a combination of climate and land use, as other catchment characteristics (e.g., topography, soil, and



lithology) usually do not change on a short term (Farjad et al., 2017; Shuster et al., 2005; Wagner et al., 2018). Vice versa, hydrology affects land use as well (Wagner and Fohrer, 2019; Wagner and Waske, 2016). So far, many efforts have been made to study the influences of the change of land use area on water quality or water balance components

(Kändler et al., 2017; Shrestha et al., 2018; Wagner et al., 2016). The effects on water quality have been a concern since the 1970s (Johnson et al., 1997). Land use patterns can alter surface roughness, evapotranspiration, soil infiltration, and the interaction between surface and subsurface water (Fiener et al., 2011; Wei et al., 2007). Consequently, the amount of water and the level of carried particles, chemicals, or metals transported can be promoted or hindered, altering water quantity and quality. The effects of land use changes on catchment water resources are manifold, e.g.,

urbanization results in a significant increase in surface runoff and water yield (Ayivi and Jha, 2018), expansion of farmland area poses increased risks to non-point source pollution of nitrogen (N) and phosphorus (P) as well as soil erosion (Hacisalihoglu, 2007; Jia et al., 2013; Rajaei et al., 2017; Roberts and Prince, 2010), whereas more semi-natural vegetation (e.g., forest, bushland, or grassland) increases the ability of filtering pollutants and intercepting rainfall thus reducing water pollution and streamflow (Moreno‑Mateos et al., 2008; Yan et al., 2013). It is of great practical

importance to identify key land use changes impacting water resources, in order to achieve an effective water and land use management in a particular catchment. Changes of both the composition and spatial structure of landscape can exert diverse influences on catchment hydrology and ecological systems (Allan, 2004; Ding et al., 2016; Haidary et al., 2013; Shawul et al., 2019). It is imperative to discriminate the effects of different aspects of a certain land use class to target sustainable and comprehensive land and water management (Liu et al., 2012; Shi et al., 2013).

Earlier studies have generally measured relationships between land use transition and water quantity and quality, using the lumped indicators of landscape composition, e.g., land use proportion of the catchment area (Narain et al., 1998; Tong and Chen, 2002). However, composition indicators are rather coarse to depict the relationships, because they do not convey any details with respect to spatial settings of landscape patterns. Spatial configurations in landscapes, including the metrics of dominance, diversity, shape, cluster, and interconnection of land use patches, play a critical

part in determining the energy and matter fluxes of e.g., solar radiation, temperature, evapotranspiration, runoff, nutrients, and sediments from the landscape ecology perspective (Amiri and Nakane, 2009; Forman, 1995; Lei et al., 2019; Wu and Lu, 2021). They therefore affect hydrological and ecological processes. With the availability of advanced spatial analysis (e.g., GIS) and remote sensing techniques (RS), various landscape metrics can be acquired efficiently for an overall assessment of landscape structure, based on classified land use maps from satellite data. Landscape

metrics are sometimes more important as descriptors of water quality than composition metrics: Ding et al. (2016) found that water quality is more significantly affected by the configuration i.e. patch density (PD) or largest patch index (LPI) than by composition of the land use type in a low-order streams dominated catchment (drainage area: 35,340 km$^2$) in southeastern China. Gémesi et al. (2011) indicated that contagion, cohesion, and aggregation indices are more important than composition variables with regard to the variability in TN and TP in the Mississippi–

Atchafalaya River watershed in USA. Recent studies on land use effects on water quantity mainly focus on land use percent, rarely on landscape metrics (Anand et al., 2018; Shrestha et al., 2018). However, metrics like landscape shape, dominance, or connectivity may play critical roles in altering the hydrological cycle, e.g., fragmented forest patches



closely relate to the capacity of infiltration and interception of rainfall (Ghimire et al., 2017); hardness and straightness of land patches of farmland, urban, and natural land uses influence flow rates at different magnitudes and directions (Riitters, 2019; Shi et al., 2013); more concentrated grassland patches result in greater evapotranspiration (Yu et al., 2020). Therefore, it is necessary to assess influences of changes in different aspects of a land use class to better understand their impact on water resources dynamics.

While land use changes and the associated changes in landscape metrics have a great potential of influencing hydrology, soil erosion or water quality dynamics at different spatial and seasonal scales (Haidary et al., 2013; Jones et al., 2001; Kändler et al., 2017), some landscape metrics have a high probability for collinearity. The collinear landscape metrics carry redundant information and are not independent predictor variables (Hargis et al., 1998). They can therefore result in biased or even misleading results when using conventional multivariate regression techniques like ordinary least-square regression, particularly in the case of a small number of observations (Shawul et al., 2019; Shi et al., 2013). Compared to ordinary multivariate statistical methods which present relatively low robustness dealing with multi-collinear variables, partial least squares regression analysis (PLSR) can overcome the limitation of multi-collinearity and achieve a robust performance by using techniques of multivariate statistical projection (Shi et al., 2013). The PLSR has widely been used to measure the "cause-effect" relationships between land use changes and water resource, based on the technique of projecting predicted and observed variables onto a new space and estimating the underlying structure between projected spaces (Ferreira et al., 2017; Shi et al., 2013; Yan et al., 2013).

The Stör River is the longest tributary of the Elbe River in the northernmost federal state of Germany, Schleswig-Holstein. Intensive agricultural activities (e.g., grazing, tillage, fertilizer, and pesticide application) are common in the catchment and increase the risk of water quality pollution (Monaghan et al., 2007). A variety of amelioration measures, e.g., tile drainage and straightening or canalizing of tributaries have been implemented in the past century to sustain agriculture productivity in lowlands. These activities brought about changes in the input and transport of nutrients and in hydrological fluxes. Meanwhile, the heterogeneity of the landscape pattern has been intensified due to artificial disturbances (Goldewijk and Ramankutty, 2004; Gu et al., 2007). We previously found significant relationships between land use patterns and water quality parameters at the landscape level in the upper Stör Catchment based on measurements (Lei et al., 2021). However, a modeling approach allows for investigating the dynamic and quantitative effects of land use changes (composition and structure) measured by separate land use types on water quality and quantity, and it is necessary for developing effective and practicable strategies of improving water quality and controlling soil erosion (Pott, 2014; Ripl et al., 1996).

To identify the key land use changes controlling the spatial and temporal variations in water quantity and quality, relationships between landscape characteristics of each land use type and water quality (represented by sediment, TP and TN) and quantity (represented by evapotranspiration, surface runoff, base flow, and water yield) are explored at the subbasin scale in the upper Stör Catchment. To this end, the hydrologic model SWAT and partial least squares regression (PLSR) are employed. The study aims at (1) calibrating and validating a catchment model for streamflow, sediment, TP, and TN loads; (2) quantifying the changes of landscape characteristics and water quality and quantity variables at the subbasin scale; (3) investigating the relationships (depicted by the contribution and influence) between LUCC and water quality and quantity dynamics at the subbasin scale.





## 2 Materials and methods

### 2.1 Study area

The rural lowland catchment of the upper Stör is the focus of this study (Figure 1). It extends from the origin of the Stör River in Willingrade to the gauge in Willenscharen (Figure 1) and is free of tidal influence. The catchment has a drainage area of approximately 462 km², with a total length of the river network of about 221 km. Its temperate climate is characterized by an average annual precipitation of 850 mm and a mean temperature of 9.4 °C between 1990 and 2019, according to the records by weather stations Neumünster and Padenstedt (DWD, 2020a). The average daily streamflow measured at the catchment outlet in Willenscharen is 5.8 $m^3$ $s^{-1}$ between 1990 and 2019, with low flows (mean value: 3.8 $m^3$ $s^{-1}$) in summer (May-October) and high flows (mean value: 7.9 $m^3$ $s^{-1}$) in winter (November-April) (LKN, 2020). Discharge occurring in the highest flow period (December-March) contributes most (around 50%) to the total annual amount of stream flow. The catchment is characterized by a flat topography, descending from nearly 60 m a.s.l. in the northeast and 85 m in the western part towards 20 m in the center and to 5-10 m in the southern part. Sandy soil (Cambisol, Gley-Podsol, Podsol) dominates the catchment, particularly in the central lowland part, while some Gley soils are mainly distributed in the east and peat soils can be found in proximity to streams and near two major wetlands (Pott and Fohrer, 2017a). The catchment is dominated by rural land use composed of arable land (36.1%) and pasture (31.3%), followed by forest (18.7%), urban areas (12.8%), and a minor fraction of water and wetland as indicated by a land use map for 2019 (Lei et al., 2021). The main cultivated crops include winter cereals (wheat, barley, and rye), corn, and rapeseed.

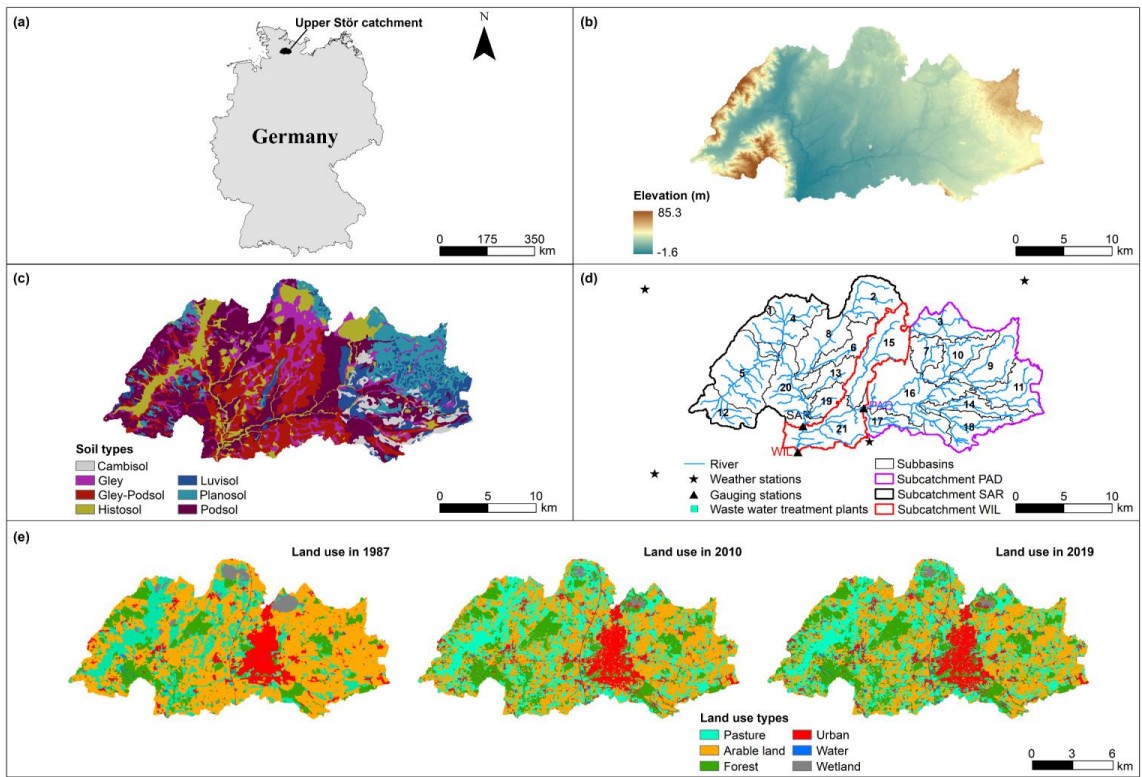

**Figure 1. Characteristics of the study area: Location of the upper Stör Catchment (a), spatial distributions of topography (b) (LvermA, 2008) and soil types (c) (Finnern, 1997), of subbasins, weather and gauging stations, and waste water treatment plants (WWTPs) (d) (Pott, 2014), as well as land use maps (e) (Lei et al., 2021; Rathjens et al., 2014; Ripl et al., 1996).**


### 2.2 Land use data and landscape metrics

Land use maps for 1987, 2010, and 2019 have been used to characterize changes in land use and landscape patterns. The earlier two maps (1987, 2010) have been adapted from Ripl et al. (1996) and Rathjens et al. (2014), respectively, and are based on Landsat TM-5 image data at 30 m resolution. The land use map for 2019 has been derived from 10 m resolution Sentinel-2 satellite images (Lei et al. 2021). The land use types are categorized uniformly as: 1) arable land (winter cereals, corn, and winter rape, and other crops), 2) pasture (meadow, field grass, and rangeland); 3) forest (deciduous and coniferous forest); 4) urban (residential, commercial and industrial areas); 5) water (rivers, ponds, and lakes) and 6) wetland (Figure 1). Water and wetland are not considered for further analysis, as they comprise only minor and mostly constant percentages.

The area percentage of land use type (PLAND) has been used as a measure of land use composition. Configuration metrics include the largest patch index (LPI), area-weighted mean shape index (AWMSI), area-weighted mean contiguity index (CONTIGAW), aggregation index (AI), and interspersion juxtaposition index (IJI), considering the dominance, shape, and interconnection of landscape (Ding et al., 2016; Gémesi et al., 2011). Composition and configuration indices of pasture, arable land, forest and urban have been selected for subsequent analysis (Table 1). They have been derived with the help of the software FRAGSTATS 4.2. All indices and their changes are analyzed at subbasin scale.

**Table 1. Description of the landscape metrics selected for the study.**

| Attributes | Metrics | Unit | Description | Abbreviation at class level | Note |
|---|---|---|---|---|---|
| Composition | Percentage of land use (PLAND) | % | Areal percentage of land use types | PLAND*a*, PLAND*p*, PLAND*f*, PLAND*u* | |
| Configuration | Largest patch index (LPI) | % | Percentage of the landscape composed of the largest patch | LPI*a*, LPI*p*, LPI*f*, LPI*u* | |
| | Area-weighted mean shape index (AWMSI) | - | The sum of the mean shape index multiplied by the area weight of each patch type involving the corresponding class | AWMSI*a*, AWMSI*p*, AWMSI*f*, AWMSI*u* | Metrics for land use type *a* (refers to arable land), *p* (refers to pasture), *f* (refers to forest), *u* (refers to urban) |
| | Aggregation index (AI) | % | Number of the same patch type being adjacent divided by the maximum number of adjacencies for the corresponding land use class | AI*a*, AI*p*, AI*f*, AI*u* | |
| | Area-weighted mean contiguity index (CONTIGAW) | - | Measure of the patch shape based on the sum of spatial connectedness multiplied by the area weight of the patch for a certain class | CONTIGAW*a*, CONTIGAW*p*, CONTIGAW*f*, CONTIGAW*u* | |
| | Interspersion juxtaposition index (IJI) | % | Measure of patch adjacency and interspersion or intermixing of patch types for a class | IJI*a*, IJI*p*, IJI*f*, IJI*u* | |

### 2.3 Hydrologic and water quality modeling

#### 2.3.1 SWAT model

The Soil and Water Assessment Tool (SWAT) is a process-based and semi-distributed eco-hydrological model with a continuous time step (Arnold et al., 1998). It is suitable for the simulation of streamflow, sediment, nutrients, and groundwater dynamics in catchments of different sizes (Aghsaei et al., 2020; Bieger et al., 2014; Haas et al., 2016; Tigabu et al., 2020). The computation of water routing, nutrient cycles and soil erosion is based on hydrologic response units (HRUs) characterized by the same land use, soil type, and slope in the same subbasin representing the spatial heterogeneity of the catchment (Arnold et al., 2013). The HRU-based calculations for the subbasins are routed through the rivers that connect the subbasins (Neitsch et al., 2011).

To accurately represent groundwater dynamics in this lowland catchment, we applied the enhanced SWAT model SWAT3s that is based on SWAT 2012 Rev. 582 (Pfannerstill et al., 2014). SWAT3s uses three groundwater aquifers and subdivides the original shallow aquifer from SWAT into a fast and a slow aquifer. SWAT3s was developed in the German lowland catchment of the Kielstau, where it better represented low flows and groundwater dynamics when compared to the original SWAT version (Pfannerstill et al., 2014). It was already successfully applied to the lowland catchment of the Treene proving its usefulness for modeling nutrients as well (Haas et al., 2017; Haas et al., 2016).





### 2.3.2 Model databases and setup

SWAT requires topography, soil, land use, hydro-meteorological input data. Topography data was obtained from a Digital Elevation Model (DEM) in 5 m resolution (LvermA, 2008) and used to delineate the watershed into 21 subbasins. Soil data and attributes for SWAT have been derived by Pott and Fohrer (2017b) from a soil type map (Finnern, 1997). The land use map for 2019 is used to build the model. Three-year crop rotations (winter wheat/winter wheat/corn; winter rape/winter wheat/corn; corn/corn/corn) are adapted from Oppelt et al. (2012) and implemented for the respective land use classes. Agriculture management

schedules and fertilization (e.g., application rates of N, P fertilizers and manure at different crop growth stages) have been determined according to the actual guidelines of agriculture practices (KTBL, 1995 and 2008; Kühling, 2011; LWK, 1991 and 2011). From the DEM a four slope classes (<1%, 1-2%, 2-5% and >5%) are defined. Slope, soil, and land use classes were combined to obtain 3618 HRUs in the catchment. The HRUs were generated without excluding any HRUs by thresholds for land use, soil, or slope class percentages, to allow for a better spatial representation. To accurately represent lowland hydrology, drainage

tiles were considered based on the estimated distribution of drained areas in the catchment (Venohr, 2000). We adapted drainage parameter values for DEP_IMP (1200 mm), DDRAIN (875 mm), TDRAIN (24 h), and GDRAIN (61 h) from a previous modeling study in the catchment (Pott and Fohrer, 2017b). Waste water treatment plants (WWTP) were implemented as point sources using data from monthly measurement campaigns in 2009 and 2010 and WWTP data vary with space and seasons (Pott, 2014). Daily values of temperature (max. and min), solar radiation, humidity, and wind speed are available from 1990 to 2019 for the climate

station Padenstedt (DWD, 2020b). Precipitation data are available for four stations (DWD, 2020b) (Figure 1). Daily streamflow is measured at the gauges in Padenstedt (PAD), Sarlhusen (SAR) and Willenscharen (WIL) from 1990 to 2019 (LKN, 2020). Daily sediment and nutrient data have been obtained during two measurement campaigns between August 2009 and August 2011 and between October 2018 and November 2019 in Willenscharen. Daily mixed samples have been taken by an automatic and cooled sampler from a depth of 0.30 m above the river bed at the central section of the stream. They have been analyzed according to

German standard procedure for water analysis (DEV) (Einheitsverfahren, 1997) in the laboratory of Department of Hydrology and Water Resources Management at Kiel University. Total suspended sediment concentration has been measured by filtering 1 l of water sample through 0.45 μm celluloseacetate filter paper and drying at 105°C. The concentration of total phosphorus (TP) has been determined by spectrophotometry, according to DEV H36 and DEV D11, while total nitrogen (TN) has been measured by chemiluminescence detection according to DEV H3. Each measurement of TP or TN concentration from unfiltered samples has

been performed based on a blank comparison analysis of distilled water and triplicate analysis of subsamples. Their concentrations have been determined by the arithmetic mean values of any two subsamples with smallest measurement differences (less than <10%).

### 2.3.3 Model calibration and validation

A step-wise calibration approach has been applied for daily streamflow (1), sediment (2), TP (3), and TN (4) data. Streamflow was

calibrated using a fifteen-year time period from 1990 to 1991 and from 2007 to 2019. The other available fifteen years (1992-2006) have been used for validation. This split of the observation data ensures an equal representation of dry, normal, and wet years in the calibration and validation period, according to the annual precipitation. First, data from the two upstream gauges Padenstedt (PAD) and Sarlhusen (SAR) have been used to calibrate parameters in the respective subcatchments (Figure 1). Then, the parameters for the area downstream of PAD and SAR and upstream of the outlet gauge Willenscharen (WIL) have been

calibrated. Sediment, TP and TN loads have been calibrated for two hydrologic years (sediment: 30/10/2009-07/08/2011; TP, TN: 08/08/2009-10/08/2011) using a model with the land use map in 2010 and validated for one hydrologic year (19/10/2018-05/11/2019) using a model with the land use map in 2019, for the entire catchment based on the daily data from Willenscharen.



The calibration has been performed based on 8000 (stream flow) and 5000 (sediment, TP, and TN loads) parameter sets generated using Latin Hypercube Sampling method (Soetaert and Petzoldt, 2010). For each parameter set a model run has been performed,

allowing for a warm-up period of 4 years. From experiences with the SWAT model in the Stör Catchment (Pott and Fohrer, 2017b) and other German lowland catchments (i.e., Kielstau and Treene catchments) (Haas et al., 2016; Lam et al., 2012; Pfannerstill et al., 2014) as well as in relevant studies from other countries (Aghsaei et al., 2020; Boongaling et al., 2018), the parameters most likely to affect hydrological and water quality processes have been selected and their preliminary ranges have been defined (Table 2). The final ranges of selected parameters have been determined based on the sensitivity of parameters to model outputs as derived

from 2000 trial runs following Guse et al. (2020) (Table 2). Calibration and validation have been carried out in R using the packages FME (Soetaert and Petzoldt, 2010), hydroGOF (Zambrano-Bigiarini, 2020) and zoo (Zeileis and Grothendieck, 2005).

The performances for modeling streamflow and sediment, TP and TN loads have been assessed using Nash-Sutcliffe efficiency (NSE), Kling-Gupta Efficiency (KGE), and Percent Bias (PBIAS) as proposed in Guse et al. (2014) and Moriasi et al. (2007). For an accurate representation of all phases of flow hydrograph for water quality simulation periods, the additional signature measure

RSR (Ratio of Root Mean Square Error to the Standard Deviation of the Observations) was used to calibrate the very high, high, middle, low, and very low periods (Haas et al., 2016; Zambrano-Bigiarini, 2020). For each of the three streamflow gauges, we pre-selected the parameter sets that yielded a KGE >0.75 for the streamflow calibration period. To particularly represent runoff dynamics during the periods of water quality measurements (Aug. 2009 - Aug. 2011 and Oct. 2018 - Nov. 2019) well, the mean of RSR for the five flow duration segments during these periods was assessed and the best 300 streamflow parameter sets indicated

by a low RSR were selected. From these 300 sets, the final parameter set yielding the highest KGE in these periods was selected. For sediment, TP, and TN calibration the parameter set that yielded the highest NSE during the calibration period was selected to represent peak loads and their dynamics well.

**Table 2. Parameters used to calibrate streamflow, sediment, total phosphorus and total nitrogen.**

| Parameters | Definition | Calibrated range | | | Calibrated value | | |
|---|---|---|---|---|---|---|---|
| Parameters used to calibrate streamflow | | | | | | | |
| | | WILL | SAR | PAD | WILL | SAR | PAD |
| r_SURLAG | Surface runoff lag coefficient | 0.1-0.6 | 0.1-0.6 | 0.1-0.5 | 0.13 | 0.13 | 0.13 |
| r_GWDELAY$fs$ | Groundwater delay time – fast shallow aquifer (days) | 48-85 | 40-80 | 65-100 | 83 | 42 | 71 |
| r_ALPHABF$fsh$ | Baseflow alpha factor – fast shallow aquifer (day$^{-1}$) | 0.17-0.38 | 0.18-0.38 | 0.05-0.22 | 0.18 | 0.26 | 0.08 |
| r_RCHRG$ssh$ | Aquifer percolation fraction – slow shallow aquifer | 0.8-0.94 | 0.08-0.58 | 0.38-0.7 | 0.91 | 0.34 | 0.44 |
| r_GWDELAY$ss$ | Groundwater delay time – slow shallow aquifer (days) | 68-105 | 58-100 | 80-120 | 80 | 92 | 87 |
| r_ALPHABF$ssh$ | Baseflow alpha factor – slow shallow aquifer (day$^{-1}$) | 0.0009-0.002 | 0.001-0.007 | 0.003-0.009 | 0.0019 | 0.0036 | 0.0064 |
| r_RCHRG$dp$ | Aquifer percolation fraction inactive deep aquifer | 0.02-0.15 | 0.015-0.14 | 0.1-0.45 | 0.14 | 0.03 | 0.15 |
| r_ESCO | Soil evaporation compensation factor | 0.85-0.98 | 0.93-1 | 0.7-0.95 | 0.86 | 0.94 | 0.77 |
| r_EPCO | Plant uptake compensation factor | 0.01-0.025 | 0.05-0.22 | 0.1-0.35 | 0.02 | 0.06 | 0.23 |
| as_CN2 | Initial SCS runoff curve number for moisture condition II | -13 - -1 | -12 - -1 | -12 - -2 | -5.64 | -3.27 | -4.89 |
| as_SOL_AWC | Available water capacity of the soil layer (mm) | -0.06 - 0.02 | -0.06 - -0.01 | -0.04 - 0.03 | -0.006 | -0.020 | 0.001 |
| m_SOL_K | Saturated hydraulic conductivity (mm h$^{-1}$) | 0.7-1.3 | 0.8-1.2 | 0.8-1.2 | 1.052 | 0.811 | 1.079 |
| Parameters used to calibrate sediment | | | | | | | |
| r_ADJ_PKR | Peak rate adjustment factor for sediment routing in the main channel | 0.55-2 | | | 0.61 | | |
| r_CH_COV_1 | Channel erodibility factor | 0.1-0.5 | | | 0.41 | | |
| r_CH_COV_2 | Channel cover factor | 0.4-0.7 | | | 0.57 | | |
| r_USLE_P | USLE support practice factor | 0.5-1 | | | 0.93 | | |
| m_SLSUBBSN | Average slope length (m) | 0.8-1.08 | | | 0.88 | | |
| m_HRUSLP | Average slope stepness (m m$^{-1}$) | 0.95-1.28 | | | 1.1 | | |
| r_LAT_SED | Sediment concentration in lateral and groundwater flow (mg l$^{-1}$) | 55-140 | | | 110 | | |
| r_USLE_K | Soil erodibility (K) factor | 0.06-0.2 | | | 0.09 | | |
| as_SOL_Z | Depth from soil surface to bottom of layer (mm) | -70-20 | | | -65 | | |
| r_USLE_C | Minimum value of USLE C factor for land cover/plant | 0.08-0.43 (cropland); 0.002-0.017 (pasture) | | | 0.192 (cropland), 0.015 (pasture) | | |
| Parameters used to calibrate total phosphorus | | | | | | | |
| r_P_UPDIS | Phosphorus uptake distribution parameter | 30-100 | | | 73.61 | | |
| r_PPERCO | Phosphorus percolation coefficient | 10-16 | | | 10.3 | | |
| r_PHOSKD | Phosphorus soil partitioning coefficient | 115-190 | | | 181.14 | | |





| r_PSP | Phosphorus sorption coefficient | 0.01-0.5 | 0.21 |
|---|---|---|---|
| r_ERORGP | Organic P enrichment ratio | 0.8-4.8 | 2.38 |
| r_GWSOLP | Concentration of soluble phosphorus in groundwater contribution to stream flow from the subbasin | 0.04-0.4 | 0.19 |
| r_SOL_SOLP | Soluble phosphorus concentration in the soil layer (mg kg$^{-1}$) | 30-90 | 32.1 |
| | | | |
| Parameters used to calibrate total nitrogen | | | |
| r_CMN | Rate factor for humus mineralization of active organic nitrogen | 0.001-0.003 | 0.002 |
| r_RCN | Concentration of nitrogen in rainfall (mg l$^{-1}$) | 1.3-6 | 5 |
| r_CDN | Denitrification exponential rate coefficient | 0.09-0.18 | 0.16 |
| r_N_UPDIS | Nitrogen uptake distribution parameter | 20-90 | 69.05 |
| r_NPERCO | Nitrogen percolation coefficient | 0.03-0.5 | 0.06 |
| r_SDNCO | Denitrification threshold water content | 0.3-0.95 | 0.95 |
| r_HLIFENGW*fsh* | Half-life of nitrate in fast shallow aquifer (days) | 30-125 | 52 |
| r_HLIFENGW*ssh* | Half-life of nitrate in slow aquifer (days) | 250-480 | 454 |
| r_SHALLSTN*ssh* | Initial concentration of nitrate in slow aquifer (mg l$^{-1}$) | 30-85 | 37.41 |

**Note:** for calibration, the parameter values were varied by replacing (r), multiplication (m) or addition/subtraction (as)

### 2.3.4 Model application

The model has been run for each of the three land use maps (in 1987, 2010, and 2019) from 1990 to 2019. As agriculture in 1987 was generally classified, it has been split as corn (12%), rapeseed (29%), and wheat (59%) randomly distributed in the catchment in SWAT model, according to the statistical data from Schleswig-Holstein Statistical Office (1992-2012). All other inputs i.e. DEM, soil data, weather data, waste water quality data, management practices, and fertilization have been kept constant, and the calibrated parameters have been adapted. Hence, each model run is performed under a different land use scenario defined by one of the three land use maps. The results from these model runs have been used to explore the influences of land use changes (LUCC) on actual evapotranspiration (ET), surface runoff (SQ), base flow (BF), and water yield (WYLD) as well as on sediment (SED), TP, and TN loads. Based on the model results, the contributions of LUCC on changes in ET, SQ, BF, and WYLD as well as SED, TP, TN at the subbasin scale are evaluated, and key impacts from LUCC are identified.

### 2.4 Partial least squares regression

Combining the features of principal component and multiple linear regression analyses, partial least squares regression (PLSR) is a robust multivariate analysis method even when dealing with multi-collinear predictor variables. The principle of PLSR is to extract a few latent components from original predictor variables that carry as much variation as possible, and which are meanwhile most likely to predict the variation in the response variable. Detailed information on the underlying theory and algorithms of PLSR is available in Abdi (2010).

In this study, PLSR is used to reveal the contribution of changes in land use types on the variation in ET, SQ, BF, WYLD, SED, TP, and TN across three time steps (1987, 2010, and 2019). The predictor variables are the changes in area percent and landscape metrics of four main land use types (arable land, pasture, forest, and urban). The response variables include the respective changes in the mean annual values of ET, SQ, BF, WYLD and SED, TP, and TN loads at the subbasin scale between 1987, 2010, and 2019. PLSR models for all of these response variables were constructed. A cross-validation is performed with 50 random repetitions on 10 equal segments of the data set. It is used to determine the number of optimal components of the PLSR model to obtain a desirable balance between the explained variation in the response ($R^2$) and predictive power of the model (measured as cross-validated goodness of the prediction: $Q^2$). The cumulative predictive ability (cumulative goodness of prediction: $Q^2_{cum}$) and the cross-validated root mean squared error (RMSECV) as the difference between actual and predicted values, are determined for each model (Yan et al., 2013). The regression coefficients (RCs) signify the direction and extent of the effect of LUCC predictor variables. The variable importance for the projection (VIP) quantifies the importance of the predictors. By Wold's criterion, a predictor with VIP<0.8 is assessed as less important (Boongaling et al., 2018; Wold et al., 2001). To achieve model parsimony, the following PLSR modeling





procedures has been conducted: First, an initial simulation of PLSR is run using all predictors. Next, new PLSR models are run by iteratively excluding the predictor with small variable importance (VIP) until the modeling procedure resulted in acceptable

variable importance or only two predictors remained. The number of components of candidate PLSR model was determined so that the $Q^2_{cum}$ is maximized (Shi et al., 2013).

All the PLSR analyses were performed with the R packages pls (Mevik et al., 2020) and mdatools (Kucheryavskiy, 2020).

## 3 Results and discussion

### 3.1 Calibration and validation of streamflow and water quality

The simulated and measured daily values of streamflow (Figure 2) and water quality (Figure 3) data are visually compared for the calibration and validation periods, and the statistical performance of the models is assessed (Table 3). The model obtains a NSE of 0.76-0.81 and a KGE of 0.82-0.85 for streamflow at the two upstream gauges Padenstedt and Sarlhusen, and a slightly higher NSE (calibration: 0.79, validation: 0.79) and KGE (calibration: 0.88, validation: 0.87) for streamflow at the outlet in Willenscharen. The PBIAS values are within the range of -2.2% - 10.6%. These values indicate a good to very good model performance for

depicting daily streamflow in the catchment (Moriasi et al., 2007). Likewise, the model shows a nearly good to very good performance for daily TN load indicated by an NSE of 0.64 for calibration and of 0.86 for validation and by a KGE ≥ 0.71 (for calibration: 0.71; for validation: 0.91), while absolute values of PBIAS are below 15%. For sediment and TP the model shows a lower performance. Sediment achieves a satisfactory performance during calibration (NSE = 0.54, KGE = 0.58, PBIAS = 12%) and a good performance during the validation period (NSE = 0.65, KGE = 0.59, PBIAS = -22.2%). For TP the model obtains a

satisfactory performance for calibration (NSE =0.56) but an unsatisfactory performance (NSE =0.29) for validation. The worse TP model performance may be due to the short and possibly different conditions during calibration and validation periods. Nevertheless, PBIAS for TP model is still within the acceptable performance range (±40 ≤ PBIAS < ±70) (Moriasi et al., 2007). It should be noted that the performance ranges from Moriasi et al. (2007) refer to a monthly time step, whereas we used a daily time step, a finer temporal scale, on which it is usually harder to achieve a good model representation. We therefore conclude that even

for daily TP the model performance is acceptable, particularly with regard to the study purpose of analyzing long-term changes in the water and matter balance.

Overall, modeled and measured daily values show clear consistency in their dynamics (Figure 2 and 3). Differences appear for a few peak flows in winter or low flow periods in summer. As already indicated by the goodness of fit measures (Table 3), the modeled streamflow matches the measured values most of the time from 1990 to 2019. However, a few single flood peaks are

underestimated in winter, e.g. on 27-28/Feb/2002, 5-6/Jan/2012, and 24-25/Dec/2014. This might be related to an insufficient representation of snow in the model, or deficiencies in single-event flood routing (Lam et al., 2012). The underestimation of peak streamflow in winter was also observed in other rural lowland catchments of Treene (Haas et al., 2016) and Kielstau (Lam et al., 2010) in northern Germany. Sediment loads are overestimated during the calibration period and slightly underestimated during the validation period mainly for a few peak values. The incorrect estimation might be due to the fact that river sediment load is also

influenced by tile drains and bank erosion in lowland catchments (Kiesel et al., 2009), while SWAT takes into account sheet erosion. A few sediment peaks in early March 2010, mid-Jan 2011 and mid-Feb 2019 are underestimated but other peaks e.g. in Nov, Dec 2009, and Mar 2019 are very well depicted. A similar behavior can be observed for TP load during the calibration and validation periods, with slight overestimation of TP in summer (April - June of 2009 and 2019) and underestimation of a few peaks in winter (between November and March). TN is generally well represented, except for only a few underestimations of extreme





peaks in winter (e.g., early March or November 2010, mid-March 2019). Overall, the underestimation of some peak loads of sediment, TP and TN might be attributed to the underestimation of corresponding peak flows.

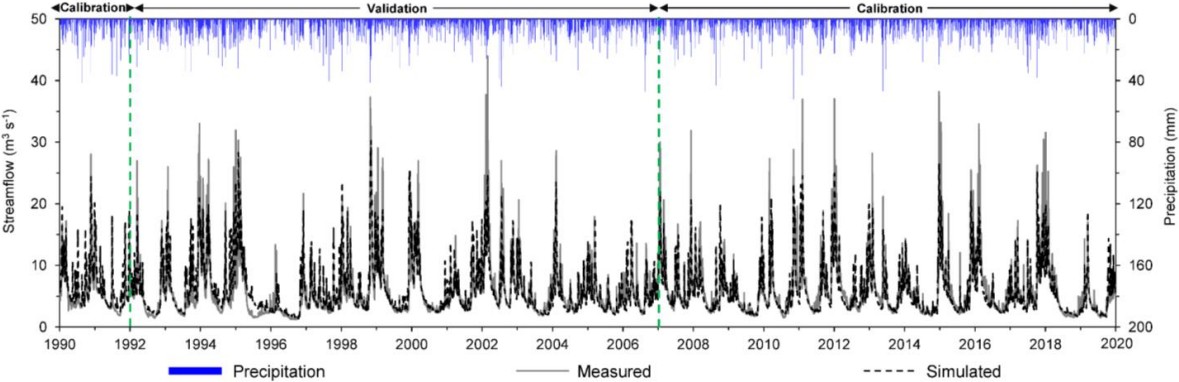

**Figure 2. Comparison of measured and modeled daily streamflow during the calibration and validation periods in Willenscharen.**

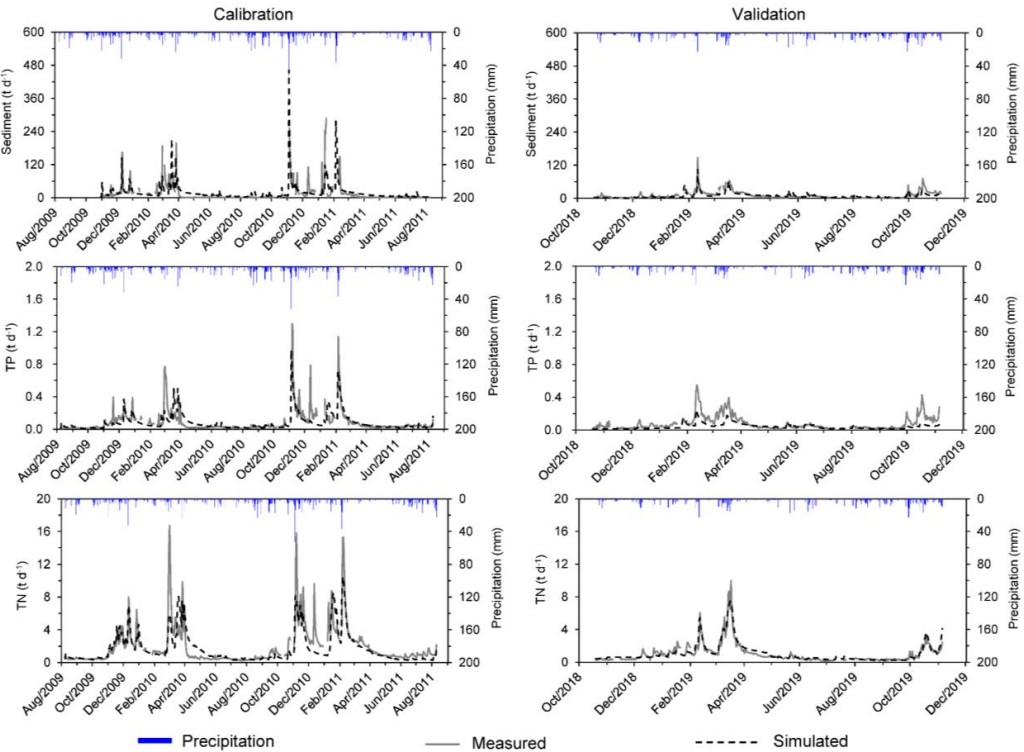

**Figure 3. Comparisons between measured and modeled daily loads of sediment, total phosphorus (TP), and total nitrogen (TN), respectively for calibration (left) and validation (right) periods**

**Table 3. Performance metrics for the model calibration and validation.**

| Index | Calibration | | | | Validation | | | |
|---|---|---|---|---|---|---|---|---|
| | Streamflow (PAD/SAR/WIL) | Sediment load | TP load | TN load | Streamflow (PAD/SAR/WIL) | Sediment yield | TP load | TN load |
| Period | 1990-1991; 2007-2019 | 30/10/2009-7/8/2011 | 8/8/2009-10/8/2011 | 8/8/2009-10/8/2011 | 1992-2006 | 19/10/2018-5/11/2019 | 19/10/2018-5/11/2019 | 19/10/2018-5/11/2019 |
| KGE | 0.85/0.82/0.88 | 0.58 | 0.65 | 0.71 | 0.84/0.85/0.87 | 0.59 | 0.22 | 0.91 |
| NSE | 0.76/0.78/0.79 | 0.54 | 0.56 | 0.64 | 0.81/0.81/0.79 | 0.65 | 0.29 | 0.86 |
| PBIAS (%) | 5.6/-2.2/0.3 | 12 | -4.7 | -11.5 | 0.7/10.6/7.2 | -22.2 | -46.2 | 5 |



### 3.2 Characteristics of land use change

Land use changes between 1987 and 2019 vary across the catchment (Figure 4). This is indicated by the individual dynamics in
the four main land use types of arable land, pasture, forest, and settlement area. Arable land has been decreasing and primarily
replaced by pasture (by 16.2% of the catchment, dark cyan in Figure 4). The decrease of arable land is more pronounced in the
northeast (e.g., subbasins 3 and 9-11) than in the northwestern part (e.g., subbasins 2, 4, 6, 8) where pasture was sometimes
converted to arable land (dark pink, Figure 4). Conversely, pasture shows an increasing trend over the period of observation. The
increase is stronger in the east as compared to the west of the catchment (Figure 4 and 5). The change of pasture is in part associated
with the stream restoration including stabilizing river shore and increasing riparian vegetation (Dickhaut, 2005; Gessner et al.,
2010). Besides, agricultural grasses may have been included in the pasture class due to the classification approach. Forest also
shows an increasing trend as indicated by green colors in Figure 4, with a strong increase in the lowlands of the middle (subbasins
6 and 13) and southern parts (subbasin 17, Figure 5). Urban area has expanded mainly around the city of Neumünster (subbasin
15 and 17) (Figure 5).

305 In addition, the subbasin-scale land use metrics varied substantially between 1987, 2010, and 2019 (Figure 6). The mean area
percent (PLAND) per subbasin declined for arable land (APLAND) by 16% and 3% during the periods of 1987-2010 and 2010-
2019, respectively. In contrast, subbasin-averaged pasture (PPLAND) increased for the period of 1987-2010 by 12% but decreased
slightly from 2010 to 2019 by 0.8%. Both forest (FPLAND) and urban (UPLAND) areas have steadily increased from 1987 over
2010 to 2019. Similar trends are found in the metrics of the percentage of largest patch index (LPI) and the interspersion
juxtaposition index (IJI). The subbasin average of LPI for arable land has decreased by 20% from 1987 to 2019, whereas the LPIs
of other land use types shows a slight and stable increase. The IJI of arable land displays an overall slight increase from 1987 to
2019, while the IJI values of other land uses have steadily and notably increased (with a net increase up to over 20%). Both the
area-weighted mean contiguity (CONTIGAW) and aggregation (AI) of each land use type have decreased over time, whereas the
area-weighted mean shape index (AWMSI) has continuously and slightly increased. Despite similar changing directions of the
land use patterns in the periods of 1987-2010 and 2010-2019, land use has been subject to more alterations in the former period
than in the latter. Additionally, CONTIGAW, AI, and IJI of arable land exhibited opposite trends in the two periods, with a decrease
from 1987 to 2010, and a slight increase from 2010 to 2019.

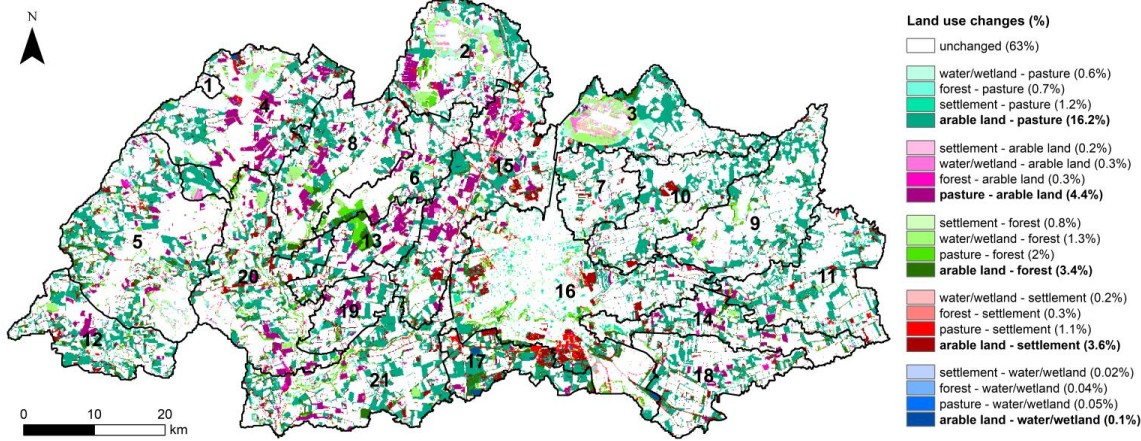

**Figure 4. Spatial distribution of land use changes between 1987 and 2019 in the Stör Catchment. Individual land use change types are**
**marked by different colors. The percentage of each change type calculated as percentage of the catchment area is given in the parentheses.**
**The strongest change is marked in bold.**





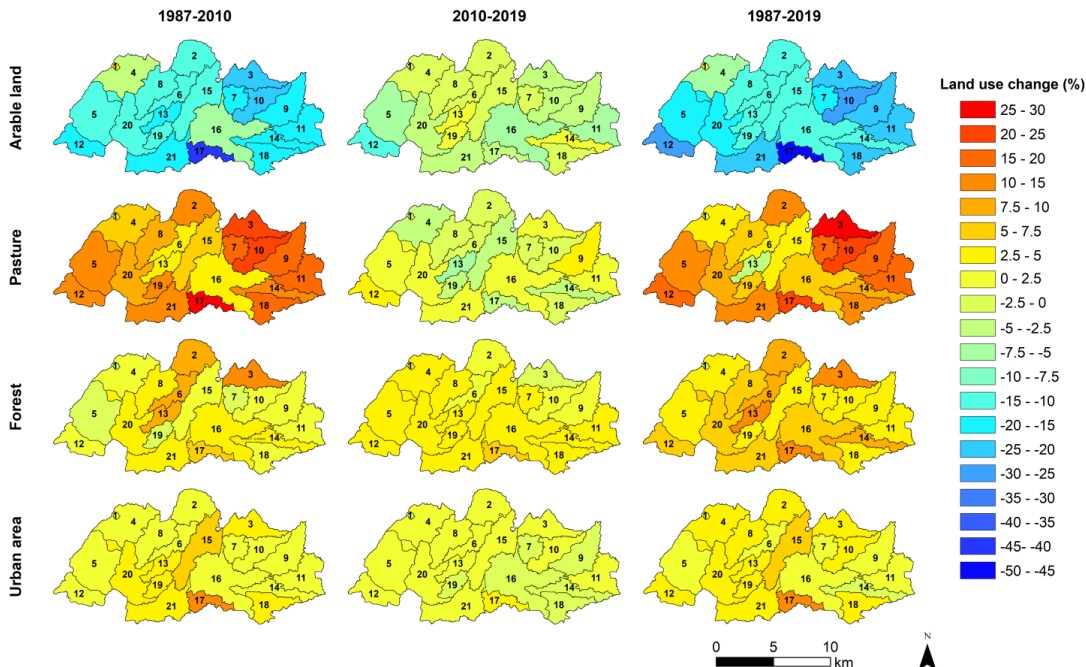

**Figure 5. Spatial distribution patterns of the change of each land use type between 1987, 2010, and 2019.**

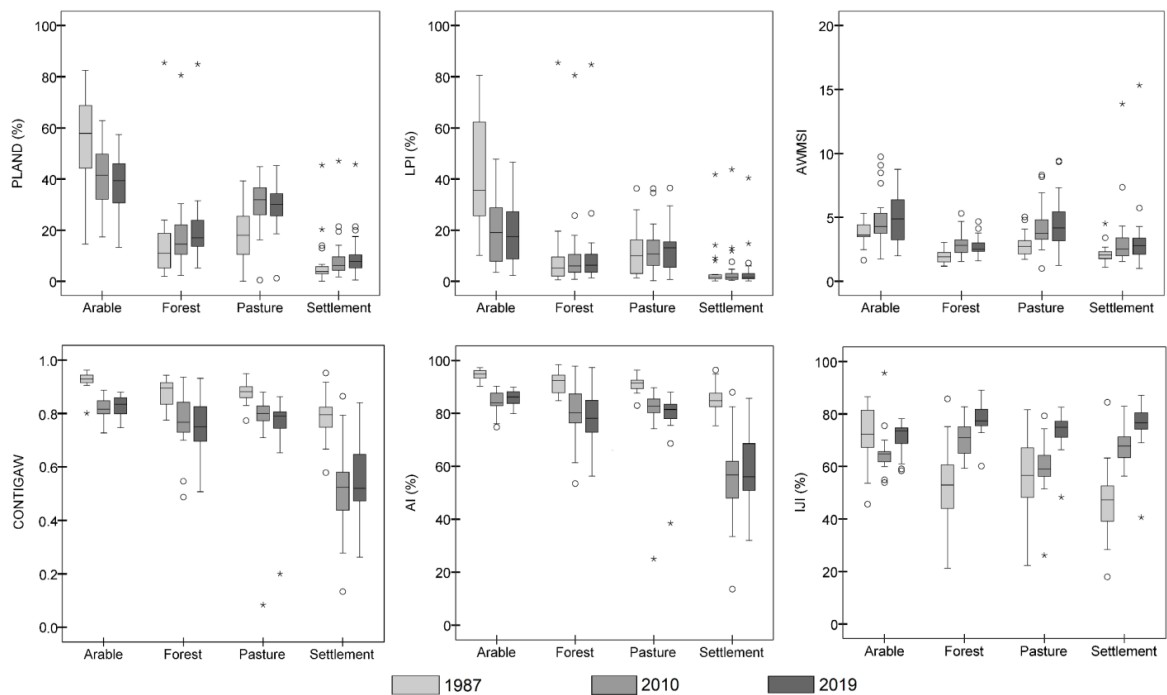


**Figure 6. Changes of land use metrics between 1987, 2010, and 2019 in the Stör Catchment.**





### 3.3 Differences of changes in water quantity and quality

Using the results from the three different model runs based on three land use maps of 1987, 2010, and 2019, we calculated changes in water quantity and quality. The spatial distribution of the variations in modeled subbasin-scale actual evapotranspiration (ET),

surface runoff (SQ), base flow (BF), water yield (WYLD), and loads of sediment (SED), total phosphorus (TP), and total nitrogen (TN) between 1987, 2010, and 2019 are shown in Figure 7. ET and SQ are mostly characterized by increases of up to 10.8 mm and 11.4 mm, respectively from 1987 to 2019, with slight decreases by up to 3.8 mm in several subbasins between 2010 and 2019. The most significant increase in ET occurs in subbasins which show a larger increase in forest from 1987 to 2019, such as subbasins 8, 12 and 17 (Figure 5). SQ shows a stronger increase in the middle-western subbasins which experienced larger expansion of

urban (Figure 5), with the strongest increase of SQ occurring in subbasins 15 and 17 that experienced the largest increase of urban area between 1987 and 2019. This might be attributed to the increased surface sealing (Anand et al., 2018; Sood et al., 2021). Contrarily, BF and WYLD have decreased by up to 20 mm and 13 mm, respectively in most subbasins in the periods 1987-2010 and 1987-2019, with slight increases between 2010 and 2019. The loads of SED, TP, and TN show notable decreasing trends from 1987 to 2019. Pronounced reductions of SED (7.8-18.2 t km$^{-2}$) occur in the relatively steeper northeastern corner (e.g., subbasins

3, 9-10) and the southwestern corner (e.g., subbasins 5 and 12) and subbasin 17, while the decrease is weaker in the mid-west. Overall, the changes in TP and TN loads show a weak decrease in the (mid) west and more pronounced decreases in the east and steeper southwest of the catchment (Figure 7). The most pronounced net decrease of TP and TN loads are observed in subbasins 12 and 17, corresponding to the largest decrease of arable land percentage (50% in subbasin 17, 30% in subbasin 12) between 1987 and 2019. The single subbasin that has experienced a slight increase of sediment or TP load is subbasin 1, which is characterized

by the least reduction of arable land and minor decrease of forest. The most significant decrease in nutrients and sediment has occurred in subbasins which have experienced notable increases of pasture or forest and a decrease of arable land, e.g., subbasins 12 and 17 (Figure 5). Overall, variations in surface runoff, sediment, TP, and TN are depicted by spatially explicit patterns on the subbasin scale. It is necessary to consider this spatial heterogeneity, when establishing management measures in order to improve water quality.



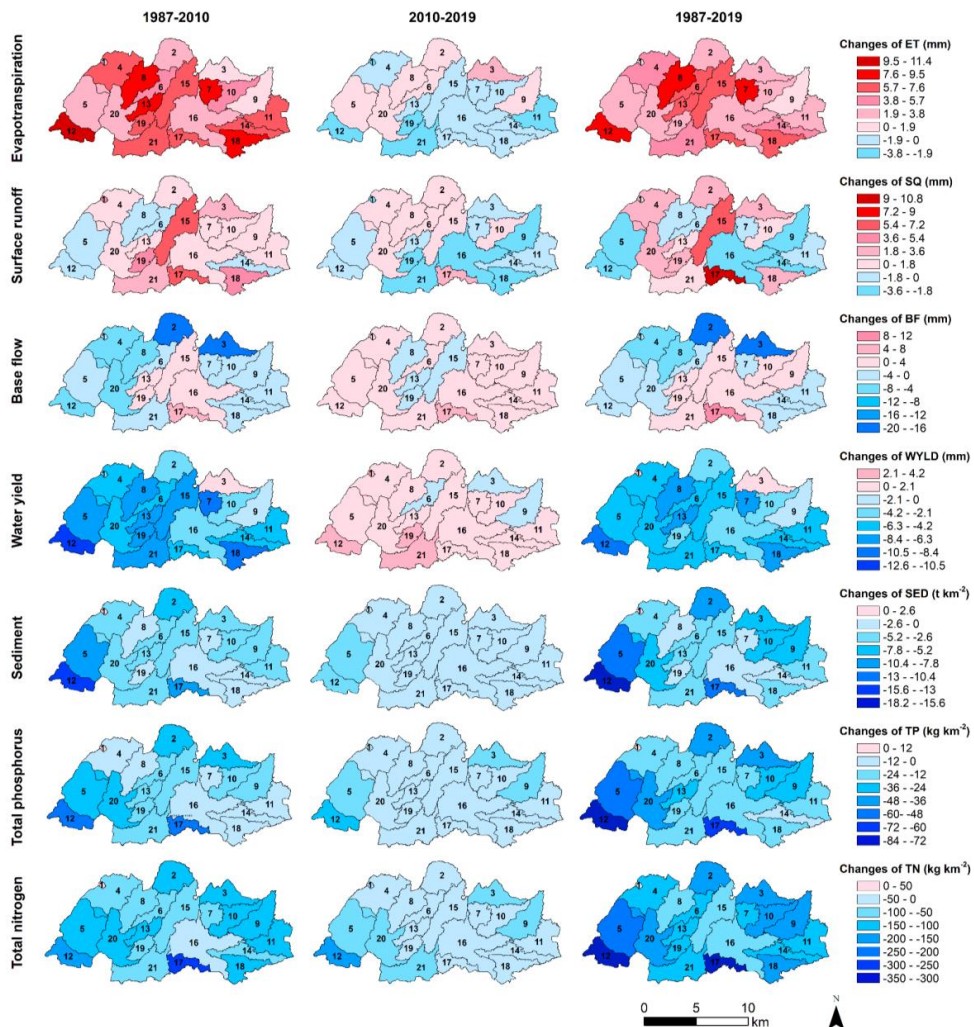


**Figure 7. Spatial distribution of variations in water quantity and water quality variables during the periods of 1987-2010, 2010-2019, and 1987-2019 at subbasin scale.**

**3.4 Influences of changes in land use metrics on water quantity and quality**

**3.4.1 Contributions of LUCC to variations in water quantity and quality**

A summary of the PLSR models separately constructed for ET, SQ, BF, WYLD, SED, TP and TN, is provided in Table 4. The prediction plots for the seven variables by applying the PLSR models are shown in Figure 8. The changes in water quantity and quality could be reasonably explained by the constructed PLSR models ($0.61<R^2<0.88$, $0.57<Q^2<0.85$, Table 4). The comparison of the actual and predicted values (in Figure 8) illustrates the accuracy of the model calibration and cross-validation. For the ET and WYLD models, the percentage of unexplained variation decreases with increasing number of components, whereas the

prediction error of cross-validated observations (indicated by cross-validated root mean squared error, RMSECV) is minimal with one or two components, respectively. This indicates that adding more components does not improve the correlation with the residuals of the response variables (Onderka et al., 2012). Overall, 60.5% and 68.3% of the variations in the changes in ET and WYLD can be explained by the first component and the first two components, respectively. Adding other components does not





strongly increase the cumulative explained variations (only by +4.2-5.4%) in ET and WYLD changes from 1987 to 2019 (Table
4). For SQ, two components are extracted for the PLSR model, with 58.9% of variation is explained on the first component and
cumulative explained variations increase to 81.3% when adding the second component. For all other variables, the minimum
RMSECV is achieved with models using five components. For base flow, 37.4% of the variation in the dynamics is explained by
the first component, cumulatively 64.2% adding the second component, and ultimately 87.7% with a consecutive addition of third,
fourth, and fifth component. For the changes in loads of sediment, TP, and TN, the first component of the models always explains
the majority of the variation (43.7-63%, Table 4). With all water quality variables together, approximately 75% of the variation is
accurately explained on average.

Approximately 70-80% of the variations in water quantity and quality dynamics were explained by LUCC, underlining the
importance of LUCC on catchment water resources. Better explanations (over 81%) of SQ and BF by LUCC confirmed the
significant influences of landscape heterogeneity on surface runoff and groundwater dynamics (Kändler et al., 2017; Xu et al.,
2020; Zhang and Schilling, 2006). Only a quarter of the variations in sediment, TP, or TN cannot be interpreted by LUCC, which
demonstrates that rural landscape patterns are essentially important in controlling nutrients pollution. The minor unexplained
fraction may be attributed to potential changes in waste water treatment which sometimes remained constant in our modeling
approach. Lower explanation of TP may be additionally due to the lower SWAT model performance for TP, the susceptibility of
P to soil or geomorphology properties (Maranguit et al., 2017; Noe et al., 2013). More than 60% of the variations in ET and WYLD
are explained by LUCC. The unexplained fraction may be attributed to the different contributions from specific crops (included in
SWAT) and the lumped agriculture land use class as well as the compensating effect of subbasins (Wagner et al., 2013).

**Table 4. Summary of the PLSR models of evapotranspiration (ET), surface runoff (SQ), base flow (BF), water yield (WYLD), sediment yield (SED), total phosphorus load (TP) and total nitrogen load (TN) at subbasin scale.**

| Response Variable $Y$ | $R^2$ | $Q^2$ | Component | Explained variability in $Y$ (%) | Cumulative explained variability in $Y$ (%) | RMSECV | $Q^2_{cum}$ |
|---|---|---|---|---|---|---|---|
| ET | 0.61 | 0.57 | **1** | 60.5 | 60.5 | 2.32 (mm) | 0.568 |
| | | | 2 | 2.4 | 62.9 | 2.35 (mm) | 0.558 |
| | | | 3 | 1.2 | 64.1 | 2.44 (mm) | 0.524 |
| | | | 4 | 0.2 | 64.3 | 2.41 (mm) | 0.535 |
| | | | 5 | 0.4 | 64.7 | 2.41 (mm) | 0.534 |
| SQ | 0.81 | 0.78 | 1 | 58.9 | 58.9 | 1.70 (mm) | 0.561 |
| | | | **2** | 22.4 | 81.3 | 1.20 (mm) | 0.783 |
| BF | 0.88 | 0.85 | 1 | 37.4 | 37.4 | 4.61 (mm) | 0.230 |
| | | | **2** | 26.8 | 64.2 | 3.92 (mm) | 0.442 |
| | | | **3** | 9.7 | 73.9 | 3.15 (mm) | 0.640 |
| | | | **4** | 8.8 | 82.7 | 2.59 (mm) | 0.757 |
| | | | **5** | 5.0 | 87.7 | 2.05 (mm) | 0.847 |
| WYLD | 0.68 | 0.61 | 1 | 64.6 | 64.6 | 2.43 (mm) | 0.611 |
| | | | **2** | 3.7 | 68.3 | 2.43 (mm) | 0.614 |
| | | | 3 | 0.9 | 69.2 | 2.46 (mm) | 0.602 |
| | | | 4 | 0.4 | 69.6 | 2.47 (mm) | 0.598 |
| | | | 5 | 0.4 | 70.0 | 2.49 (mm) | 0.592 |
| SED | 0.77 | 0.67 | 1 | 43.7 | 43.7 | 2.76 (t km$^{-2}$) | 0.382 |
| | | | **2** | 19.2 | 62.9 | 2.50 (t km$^{-2}$) | 0.493 |
| | | | **3** | 11.1 | 74.0 | 2.13 (t km$^{-2}$) | 0.630 |
| | | | **4** | 1.6 | 75.6 | 2.08 (t km$^{-2}$) | 0.650 |
| | | | **5** | 1.0 | 76.6 | 2.03 (t km$^{-2}$) | 0.667 |
| TP | 0.76 | 0.65 | 1 | 51.5 | 51.5 | 12.03 (kg km$^{-2}$) | 0.468 |
| | | | **2** | 10.7 | 62.2 | 11.14 (kg km$^{-2}$) | 0.544 |
| | | | **3** | 10.4 | 72.6 | 10.32 (kg km$^{-2}$) | 0.608 |
| | | | **4** | 3.0 | 75.6 | 9.80 (kg km$^{-2}$) | 0.647 |
| | | | **5** | 0.7 | 76.3 | 9.71 (kg km$^{-2}$) | 0.653 |
| TN | 0.73 | 0.68 | 1 | 63.0 | 63.0 | 43.04 (kg km$^{-2}$) | 0.597 |
| | | | **2** | 5.8 | 68.8 | 40.56 (kg km$^{-2}$) | 0.643 |
| | | | **3** | 3.1 | 72.1 | 39.20 (kg km$^{-2}$) | 0.666 |
| | | | **4** | 0.5 | 72.6 | 38.90 (kg km$^{-2}$) | 0.671 |
| | | | **5** | 0.7 | 73.3 | 38.51 (kg km$^{-2}$) | 0.678 |

**Note**: $R^2$ indicates the goodness of fit of the model; $Q^2$ indicates the cross-validated goodness of prediction; RMSECV indicates cross-validated root mean squared error; $Q^2_{cum}$ indicates the cumulative cross-validated goodness of predication over all the selected PLSR components; the components selected for each model are highlighted in bold.





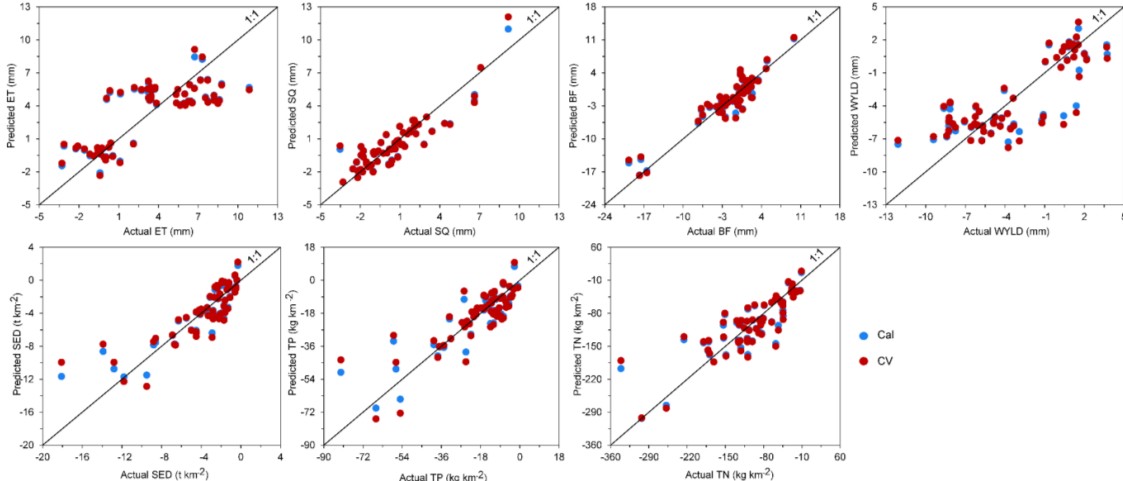

**Figure 8. Comparison of subbasin-scale changes in evapotranspiration (ET), surface runoff (SQ), base flow (BF), water yield (WYLD), sediment (SED), total phosphorus (TP), and total nitrogen (TN) as derived from the SWAT model and the predicted values from the PLSR models. The changes were obtained based on land use changes between 1987 and 2010, 2010 and 2019, and between 1987 and 2019, respectively. Cal indicates calibration. CV indicates cross validation.**

### 3.4.2 Effects of LUCC predictors on water quantity and quality

According to the PLSR results, each category of the landscape indices including percentage (PLAND), largest patch (LPI), shape (AWMSI), contiguity (CONTIGAW), aggregation (AI), or interspersion (IJI), plays an essential role in influencing as least one water quantity or quality variable (Table 5). The effects on the changes in ET, SQ, BF, WYLD, SED, TP, and TN are measured using weights, regression coefficients (RCs), and VIP values in the PLSR models. VIPs for predictors included into the models are greater than 0.8. For the ET model, the highest VIPs are obtained in predictors AI$a$ and CONTIGAW$a$ (VIP = 1.25, RCs = -0.122),

followed by PLAND$a$ (VIP = 1.037, RC = -0.101) and AI$u$ (VIP = 1.03, RC = -0.1). ET tends to decrease with larger aggregation (AI$a$) and contiguity (CONTIGAW$a$) indices, and arable land percent (PLAND$a$) (negative RCs), whereas it increases with more pasture (PLAND$p$) (positive RC). In the case of surface runoff, the first and second components of the model are dominated by PLAND$u$ on the positive side, with minor positive effect from PLAND$a$ on the second component (Table 5). The urban area percent (PLAND$u$) obtains largest VIP of 1.173, and are identified as most important influencing the change in surface runoff.

Surface runoff increases with an increase in arable (PLAND$a$) and urban areas (PLAND$u$) (RCs=0.403, 1.161, respectively). For base flow, in addition to arable land, pasture plays a key role in explaining its variation. Arable land (PLAND$a$), pasture (PLAND$p$) percent and area-weighted shape index of pasture (AWMSI$p$) obtain the largest VIPs of 1.259, 1.03, and 1.063, respectively. All show negative correlations with base flow. AI$a$ and CONTIGAM$a$ are important predictors for water yield with large VIPs of 1.226 and 1.218, respectively. Their higher values result in an increase of water yield. For sediment, TP or TN models, the selected

components are dominated by areal percentages of arable land and pasture, in addition to the landscape metrics of arable land. The models obtain the largest regression coefficients or VIPs for PLAND$a$, LPI$a$, or PLAND$p$. They have VIPs of 1.0113-1.173 for sediment, 1.089-1.305 for TP, 1.005-1.232 for TN, respectively. Inferred by the RCs, an increase in sediment, TP, or TN occurs with increasing arable land (RCs: 0.602-0.884), while a decrease may occur with higher percentage of arable land in largest patches (LPI$a$) (RCs: -0.74 - -0.225), or with more pasture area (RCs: -0.693 - -0.122).

LPI$a$, AI$a$ and CONTIGAW$a$ are the most important landscape structure indicators affecting water quantity or quality (VIP ≥1 most of the time, Table 5). AI$a$ and CONTIGAW$a$ have positive impacts on WYLD while negative impacts on ET. By definition, AI$a$ and CONTIGAW$a$ would increase, respectively, when arable landscape patches are more clumped and contiguous (Shi et al.,



2013; Uuemaa et al., 2009). Clumped and connected agriculture patches with fewer edges have reduced more infiltration of runoff, compared to small scattered patches (Boongaling et al., 2018), thus resulting in the increase of water yield amount in the catchment.

Our results also corroborate with Ayivi and Jha (2018) who reported that increased water yield and base flow occur with increasing cohesive and aggregated agriculture. Negative impacts on ET may be explained by the interactive changes between arable and pasture, i.e., arable land has been increased at the cost of losing pasture, and vice versa. The negative effect of AWMSI$p$ on base flow implies that the coarse grass landscape has a higher capacity of absorbing and intercepting rainfall thereby resulting in lower base flow. Though landscape metrics are more often used to explain water quantity than quality variables (Table 5), the negative

influences of LPI$a$ on sediment and nutrients, and positive influences of AWMSI$a$ on sediment and TP cannot be overlooked. This is in agreement with previous findings that scattered and complicated agriculture patches are susceptible to soil erosion and thus water quality deterioration (Nafi'Shehab et al., 2021; Yan et al., 2013).

The change in the percentage of arable land is most responsible for water quantity and quality dynamics, with VIP values greater than 1 for all response variables but WYLD. This may be explained by the fact that the decrease in arable land is the strongest.

The negative correlations between PLAND$a$ and evapotranspiration (ET) and base flow (BF) imply that conversion of arable land to e.g., pasture or forest would result in increased ET and BF, due to higher capability of plant evapotranspiration and slower water transmission, which is in agreement with previous findings that perennial vegetation is more likely to increase ET (Li et al., 2017; Peel et al., 2010) and the decrease in agriculture leads to increased annual base flow (Basuki et al., 2019). Changes of the percentage of arable land positively influence SQ, WYLD, SED TP, and TN loads. Less runoff interception by crops and additional runoff

routes resulting from implementation of tillage practices (e.g., tractor road) can result in increased surface runoff (SQ). The lower ET amount of crops compared to pasture and forest is in part responsible for the increase in WYLD. Soil erosion might be accelerated due to uncovered and fragile soil by tillage practices implemented in cultivated areas as well as the increased surface runoff. N and P pollution is prone to occur in arable areas, which have a high risk of generating nutrient pollutants from excessive fertilizer or manure and eroded soil particles. The positive relationships between arable land percent and SQ, WYLD, SED TP,

and TN loads are found in other studies as well (Mirghaed et al., 2018; Sood et al., 2021; Wagner et al., 2013; Wang et al., 2019). Pasture shows a positive influence on ET and negative influences on sediment, TP, and TN. This also illustrates that more grassland (or rangeland) would increase plant evapotranspiration process. Pasture can improve water quality due to reduced soil erosion and nutrient transportation rate, as well as the high uptake and infiltration of nutrients by vegetation cover (Ding et al., 2016; Hatano et al., 2005; Li et al., 2008).

By applying the quantitative results that the increases in arable or pasture areas most significantly intensify or reduce the risk of soil erosion and nutrient pollution, respectively, individual subbasins can be identified as nutrient pollution "source" or "sink". Based on these results, it is possible to develop a set of more targeted strategies to effectively control diffuse pollution at a spatial scale. At the same time, best management practices such as proper fertilization, abate of traditional tillage, crop rotation, vegetation buffer, are important to improve water quality in rural catchments (Haas et al., 2017; Pott and Fohrer, 2017a). Urban expansion is

most important influencing surface runoff, the increase in urban area percent results in an increase of it (regression coefficient value > 1.16, Table 4). Similar results have been found, e.g., by Shi et al. (2007) who discovered that increased urbanized land led to increased runoff, by increasing peak flood runoff and decreasing runoff confluence time, in a typical urbanized region (Shenzhen) in China. It is therefore necessary to increase the frequency of measuring runoff, sediment and nutrient, particularly during the course of storm flood events in settlement area. Unlike previous findings (Wang et al., 2018; Yan et al., 2013), forest

properties have not exerted significant influences, probably due to only minor temporal changes in some landscape metrics, e.g., area percent (PLAND), dominance (LPI), and shape (AWMSI) of forest (Figure 6).





**Table 5. Regression coefficients (RCs), VIP and weight values of each PLSR model.**

| Predictors | ET | | | SQ | | | | BF | | | | | | | WYLD | | | |
|---|---|---|---|---|---|---|---|---|---|---|---|---|---|---|---|---|---|---|
| | RC | VIP | W*[1] | RC | VIP | W*[1] | W*[2] | RC | VIP | W*[1] | W*[2] | W*[3] | W*[4] | W*[5] | RC | VIP | W*[1] | W*[2] |
| PLANDa | -0.101 | **1.037** | -0.017 | 0.403 | 0.790 | -0.048 | *0.189* | -1.654 | **1.259** | -0.001 | *-0.128* | *-0.135* | *-0.208* | *-0.201* | 0.043 | 0.882 | 0.017 | -0.042 |
| PLANDp | 0.089 | 0.918 | 0.015 | | | | | -1.474 | **1.030** | -0.034 | 0.024 | *-0.117* | *-0.304* | *-0.256* | 0.011 | 0.866 | -0.015 | 0.072 |
| PLANDf | | | | | | | | -0.575 | 0.915 | -0.035 | -0.074 | -0.072 | -0.045 | 0.092 | | | | |
| PLANDu | 0.080 | 0.818 | 0.013 | 1.161 | **1.173** | 0.090 | *0.173* | | | | | | | | | | | |
| LPIa | -0.088 | 0.906 | -0.015 | | | | | | | | | | | | | | | |
| AWMSIp | | | | | | | | -0.143 | **1.063** | -0.052 | -0.058 | 0.059 | 0.093 | -0.013 | | | | |
| AWMSIf | 0.085 | 0.870 | 0.014 | | | | | | | | | | | | -0.039 | 0.837 | -0.016 | 0.041 |
| AIa | -0.122 | **1.254** | -0.020 | | | | | | | | | | | | 0.187 | **1.226** | 0.024 | 0.025 |
| AIP | -0.094 | 0.961 | -0.016 | | | | | | | | | | | | 0.100 | 0.924 | 0.018 | -0.009 |
| AIu | -0.100 | **1.030** | -0.017 | | | | | | | | | | | | 0.212 | **1.068** | 0.020 | 0.058 |
| CONTIGAWa | -0.122 | **1.251** | -0.020 | | | | | | | | | | | | 0.184 | **1.218** | 0.024 | 0.024 |
| CONTIGAWP | -0.087 | 0.891 | -0.015 | | | | | | | | | | | | 0.112 | 0.880 | 0.018 | 0.004 |
| CONTIGAWu | -0.094 | 0.959 | -0.016 | | | | | 0.281 | 0.805 | 0.040 | 0.029 | -0.078 | 0.064 | 0.011 | 0.198 | **1.007** | 0.019 | 0.054 |
| IJIa | | | | | | | | 0.038 | 0.859 | 0.040 | 0.024 | 0.098 | *-0.142* | -0.091 | | | | |

| Predictors | SED | | | | | | | TP | | | | | | | TN | | | | | | |
|---|---|---|---|---|---|---|---|---|---|---|---|---|---|---|---|---|---|---|---|---|---|
| | RC | VIP | W*[1] | W*[2] | W*[3] | W*[4] | W*[5] | RC | VIP | W*[1] | W*[2] | W*[3] | W*[4] | W*[5] | RC | VIP | W*[1] | W*[2] | W*[3] | W*[4] | W*[5] |
| PLANDa | 0.602 | **1.165** | 0.027 | 0.038 | *0.106* | 0.037 | 0.040 | 0.755 | **1.305** | 0.029 | 0.031 | *0.117* | *0.142* | 0.059 | 0.884 | **1.232** | 0.033 | *0.103* | *0.133* | *0.166* | *0.333* |
| PLANDp | -0.693 | **1.173** | -0.026 | -0.022 | *-0.124* | -0.096 | -0.099 | -0.499 | **1.089** | -0.025 | -0.007 | -0.099 | -0.074 | 0.002 | -0.122 | **1.005** | -0.030 | -0.054 | -0.049 | 0.031 | *0.324* |
| PLANDu | 0.013 | 0.908 | -0.022 | -0.033 | 0.020 | 0.097 | *0.116* | -0.045 | **1.038** | -0.025 | -0.033 | 0.005 | 0.057 | *0.137* | 0.028 | **1.013** | -0.024 | -0.032 | 0.052 | *0.197* | 0.093 |
| PLANDf | | | | | | | | -0.009 | 0.821 | -0.016 | -0.053 | 0.061 | 0.047 | 0.004 | | | | | | | |
| LPIa | -0.632 | **1.113** | 0.015 | -0.095 | *-0.117* | -0.037 | -0.070 | -0.740 | **1.205** | 0.017 | -0.064 | -0.208 | -0.091 | -0.057 | -0.225 | 0.945 | 0.023 | -0.054 | *-0.209* | 0.028 | 0.019 |
| LPIp | 0.397 | 0.819 | -0.009 | 0.075 | 0.086 | -0.043 | 0.020 | | | | | | | | | | | | | | |
| AWMSIa | 0.472 | 0.902 | 0.007 | *0.103* | -0.017 | 0.073 | 0.080 | 0.492 | 0.817 | 0.008 | 0.087 | 0.020 | 0.093 | 0.085 | | | | | | | |
| AWMSIp | -0.445 | **1.087** | -0.023 | -0.077 | -0.050 | -0.022 | *0.107* | -0.152 | 0.872 | -0.019 | -0.031 | -0.057 | *0.127* | -0.001 | | | | | | | |
| CONTIGAWa | 0.039 | 0.877 | 0.023 | -0.001 | -0.042 | -0.024 | 0.075 | 0.079 | 0.864 | 0.021 | -0.027 | -0.013 | 0.015 | 0.069 | 0.114 | 0.840 | 0.022 | -0.072 | 0.037 | 0.019 | 0.077 |
| AIa | -0.053 | 0.876 | 0.022 | -0.006 | -0.055 | -0.039 | 0.041 | 0.008 | 0.856 | 0.021 | -0.030 | -0.025 | 0.000 | 0.052 | -0.034 | 0.833 | 0.022 | -0.081 | 0.015 | -0.024 | -0.038 |

**Note:** VIP values greater than 1 were marked in bold; the absolute weights greater than 0.1 were marked in Italic.

## 4 Conclusion

In this study the separate contributions of changes in land use on the dynamics of seven water quantity and quality variables, i.e., actual evapotranspiration (ET), surface runoff (SQ), base flow (BF), water yield (WYLD), sediment (SED), total phosphorus (TP), and total nitrogen (TN) loads have been quantified by applying an integrated approach of hydrological modeling (SWAT) and partial least squares regression (PLSR). The influences of the changes in individual landscape metrics on variations in water quantity and quality have been measured and identified using a scenario analysis for three different land use maps of the past.

With an exceptional data set that covers land use changes and three water quality campaigns over a period of three decades, a hydrologic model was set up and showed reasonable performance on the daily time scale. The results of the scenario analysis indicate that the dynamics of all water quantity and quality variables are largely explained (61-68% of the variations in ET and WYLD; 75-88% of the variations in other water quantity and quality variables) by land use changes (LUCC) between 1987 and 2019. Landscape metrics show a stronger effect on water quantity than on water quality. Moreover, water quantity and quality

variables are most influenced by arable land change. The percentage (PLANDa), contiguity (CONTIGAWa), and aggregation (AIa) of arable land are identified as primary landscape metrics controlling the variations in BF, ET and WYLD. Greater percentages of settlement area and arable land may significantly accelerate runoff processes. Land planners and decision makers

probably need to control land use patterns in runoff-sensitive areas to minimize negative impacts. Sediment, TP, and TN loads are closely associated with pasture and arable land. The expansion of arable land (PLAND*a*) may exacerbate soil erosion and P and N

pollution. The arable land in large and aggregated (LPI*a*) or simpler shape (AWMSI*a*) patches can help to mitigate soil erosion and water quality deterioration. The results indicate that the smaller changes in forest did not exert significant influence on water quantity and quality.

The approach applied in this study identifies the important influences of land use changes on water quantity and quality, which are helpful for formulating an informed and targeted plan with regard to land and water resource management. This approach is

applicable to other catchments to predict both the water quality and hydrological responses to land use changes with the help of time-sequenced land use data.

**Data availability**

The datasets used in this study may be available upon request to the corresponding author.

**Author contribution**

Chaogui Lei, Paul D. Wagner, and Nicola Fohrer designed the experiments and Chaogui Lei carried them out. Chaogui Lei and Paul D. Wagner developed the model codes. Chaogui Lei performed the simulations with the supervision by co-authors. Chaogui Lei prepared the manuscript with many contributions from all co-authors.

**Competing interests**

The authors declare that they have no conflict of interest.

**Acknowledgments**

We gratefully acknowledge the funding from the China Scholarship Council (CSC) for the first author. We deeply appreciate the assistance with the field sampling and lab analysis by lab technicians Bettina Hollmann, Falko Torreck, Monika Westphal, and Imke Meyer. Special thanks go to Cristiano Andre Pott for collecting water quality data for 2009-2011 and to our students Anne-Kathrin Wendell, Henrike Risch, Jia Yuan, Josephine Loeck, Lisa Jensen, Marian Scheffler, and Tanja Boehlke for supporting the

water quality measurement campaign in 2018-2019.

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
