# Peer review of "Influences of land use changes on the dynamics of water quantity and quality in the German lowland catchment of the Stör"

_Hydrology and Earth System Sciences, 2021_

## Author Response (AR1)

*We thank the reviewers and editor very much for your valuable comments and suggestions. Please find our response (in Italic) started with "Re:" under each comment below.*

**Response to reviewers' comments**

**First reviewer**

This paper shows a comprehensive investigation of the influences of land-use changes on the dynamics of water quality and quality using SWAT. The manuscript Yet, I have some concerns.

1. My biggest concern is if the methodology you applied here fits the study's purposes. Specifically, the model is set up, calibrated, and validated using the land use map of 2019 as a static input. Then the authors applied the land use maps in 1987 and 2010 as scenarios with the parameter setting calibrated with the map of 2019 as input. The land-use maps are inherent catchment characteristics. Moreover, the years of the land use maps are within the simulation period. From my perspective, they are not scenarios. Therefore, they should be all used as forcing in the model set up and calibration, e.g., as in Anand et al. (2018). By doing so, the authors can still do the same calculations to investigate the influences of land-use changes on the water quantity and quality dynamics. Please justify the reason for using the land use map of 2019 as a static input for model set up and calibration instead of forcing with all land use information that you have.

*Re: We thank the reviewer very much for this general comment and gladly provided reasoning for the used methodology.*

*We agree with the reviewer that a calibration using one static land use map for streamflow was applied. We chose to use the map of 2019 as it provides the most detailed land use classes and is closer to the two periods for which water quality data was available. Moreover, land use is not as static as it seems, as three-year crop rotations have been implemented indicating that various crops are rotated (changing) over time. With regards to water quality, the SWAT model is*

*calibrated for water quality data in 2009-2011 with the land use map of 2010 and validated for measured data in 2018-2019 with the land use map of 2019.*

*While we agree that a dynamic implementation of land use changes would be necessary to represent dynamic changes, in this study the model is used to compare the averaged output from three static model runs to the associated changes in land use. We therefore believe that a calibration using a static land use is suitable. We however evaluated the performance of the model with regard to streamflow for the other two land use maps and showed that the parameters derived during the current calibration are also suitable for these land use maps (using land use map 1987, NSE: 0.75 - 0.80, KGE: 0.82 - 0.88, PBIAS: -1.1 - 12.2; using land use map 2010, NSE: 0.76 - 0.81, KGE: 0.82 - 0.88, PBIAS: -2.5 - 10.3).*

2. The predictor and response variables for the PLSR are not well described. I suggest using a table to summarise or equations to describe the changes. How the changes are calculated is not clear.

*Re: We thank you so much for this helpful comment. We provided additional information about the description and calculation of the changes in the Text S2 in the supplementary materials. Besides, we also explained in detail how these changes of land use and water resources components are employed as predictors and response variables of PLSR model, respectively, in the Text S3 in the supplementary materials.*

3. The calibration processes are very difficult to follow. Please rephrase it. Is there autocalibration? If so, please specify how the objective functions are defined, and what algorithm is applied. Only giving the information of the package is not enough.

*Re: We clarified how the model has been calibrated for streamflow and water quality using a table (table 2) for describing the calibration and validation. We used an auto-calibration approach based on 8000 model runs for streamflow and 5000 model runs separately for sediment, TP, and TN. Parameter sets for these runs have been generated using Latin Hypercube Sampling. With regards to model performance assessment, a multi-metric approach was used for streamflow including KGE and RSR as objective functions. For sediment, TP, and TN, NSE was used as the objective function. We clarified this in section 2.3.3.*

*More details about the used objection functions and calculations are provided in section Text S1 in supplementary materials.*

4. In Figure 6, the notation of stars should be clarified.

*Re: The stars are used to indicate the extreme outliers. We added a legend for explaining all notations (e.g., circles, stars) in Figure 6.*

5. In Figure 7, what is the baseline of all the comparisons?

*Re: Thank you for the comment! There is not one baseline for all comparisons in Figure 7. The figure shows differences between the model runs (between 1990 and 2019) with a land use map of the first stated year and the second stated year, i.e. the first stated year in each column may be referred to as baseline (e.g. 1st column 1987-2010, baseline is 1987; 2nd column 2010-2019, baseline is 2010; 3rd column 1987-2019: baseline is 1987). We added more details in section 2.3.4 to clarify this point.*

**Second reviewer**

The study aimed at understanding the effects of land use and land cover changes on the dynamics of water quantity and quality in a plain watershed. A sophisticated watershed model and statistical techniques are employed to discern the relationships. This article includes three storylines or parts: one is about the hydrology and water quality simulation, the second is about the land use and landscape pattern and its dynamics, and the third is the relationships between the landscape settings and watershed hydrology & water quality. Each of them was complicated and can become an independent study. From this perspective, the research represents a lot of work, with additional information that the studied period has encompassed around 30 years.

However, for such a work associated with many storylines, it is challenging to balance the details and key findings. The main flaws of this study are related to these aspects. My comments are as follows:

1) The arrangement of the structure of the manuscript is inappropriate. For example, in the introduction, the authors intensively highlighted the importance

of LULC. And in most lines, the authors never mention the SWAT model, particularly on why the model is suitable for a lowland watershed. In the following text, however, the authors have intensively described the results of the SWAT model. This arrangement is confusing and would greatly weaken the reasonableness of this study.

*Re: We thank the reviewer for this very constructive comment. We agree that we can broaden the introduction to highlight the suitability of SWAT for this study. Specifically, we shortened the first and second paragraph about the importance of LULC and added a new paragraph after them, which introduces the importance and suitability of using SWAT to assess the impacts of land use changes on hydrology and water quality. We also highlight the high capability of the enhanced SWAT version (with two active groundwater aquifers) of modeling low flows of streamflow and representing groundwater process and associated water quality components in lowland catchments dominated by abundant groundwater recharge.*

2) Most of the information is redundant. For example, many of the lines were given on the model results, especially on the daily simulation. But in fact, the simulated data used in the following analysis (e.g., section 3.4) were on an annual scale. I think some of the information could be removed or at least should be moved to the supplementary materials. Thus, the manuscript could be more concise and shortened.

*Re: Thank you very much. We believe that it is important to firstly show that the model is capable of accurately representing water quality and quantity before applying it to scenario simulations. We therefore include an evaluation of the modeled results on a daily time step, which is the time scale of the constructed model. The scenario simulations were performed also based on daily time step, but we output the scenario results on the annual scale for the following analysis (e.g., in section 3.4). As the reviewer suggested, to make the manuscript more concise, we removed some information and shortened the section 3.3 on model performance assessment.*

3) The descriptions of model calibration and verification are unclear. For example, the author has collected land use data for three calendar years. How

these data were used in the simulation? Instead of lengthy and confusing descriptions, maybe use a simple table to summarize the data (or other details) used in the SWAT simulations?

*Re: We used the land use map of 2019 (detailed land use classes) for the SWAT calibration and validation for streamflow. For water quality, the land use map in 2010 was used for calibrating sediment, TP, and TN, while land use map in 2019 was used for validating sediment, TP, and TN. After all streamflow and water quality variables were well calibrated and validated, the parameters from the corresponding best model runs were applied to the three scenario simulations. Each scenario simulation was respectively performed under land use map in 1987, 2010, or 2019, during the period 1990-2019.*

*Following the reviewer's and the editor's comment, we restructured the entire section 2.3.3 to provide more clarity: 1) we explain the stepwise procedures, 2) the used parameter ranges, 3) parameter sampling and model evaluation, 4) streamflow calibration, 5) calibration of sediment, total phosphorous, and total nitrogen. Moreover, we summarized data and details for the calibration and validation in the newly added table (Table 2).*

4) The authors claimed that the studied watershed is characterized as lowland or plain, but I have not found any special findings related to such geographical characteristics.

*Re: Thank you very much. The main lowland characteristics were added in the sentence in current L109 "lowlands dominated by shallow groundwater tables and abundant groundwater recharge" in the revised manuscript (no track-change), and reflected by the relevant statement of flat topography of below 85 m in the section "Study area".*

*There are a few findings related to lowland characteristics in the dynamics of water quality loads (in Figure 7). E.g., due to land use change between 1987 and 2019, sediment, TP, and TN loads in the central areas with a lower topography (lowland) decreased least, compared to relatively steeper surroundings. This may be related to the more intense exchange between groundwater and surface water and a resultant higher contribution of nutrients from groundwater to stream in lowland. We followed the reviewer's suggestion, and in section 3.3 we added these statements "The spatial differences may be …to the stream in the*

lowland" in current L366-368 in revised manuscript (no track-change), to better highlight the findings of water quality related to lowland characteristics. Besides, we also found that, compared to the pronounced decrease of base flow in the eastern and western subbasins, base flow exhibits a slight increase in a few subbasins in the central part of catchment, which is probably attributed to a greater contribution of the shallow groundwater in the central lowland areas to low flows than in the steeper eastern and western areas. We also added this statement to the section (in current L360-362 in revised manuscript).

5) The sections of the abstract and conclusion need to be rewritten. The current abstract is lengthy with many details and lacks a summary. The conclusion part is similar, it appears to me that it is just a simple repetition of the results. I would suggest that the authors carefully summarize the main findings of the results and shorten them appropriately.

*Re: Thank the reviewer for these helpful comments! We rewrote and improved abstract and conclusion by summarizing the main findings to make them more concise.*

6) It would be necessary to add some comparisons, e.g., with some hilly and plain watersheds, on the effects of land-use changes on water quality & quantity. This is expected to expand and deepen readers' understanding of the land use effects of the watershed being studied.

*Re: We thank you very much for this constructive comment! In the discussion of results in section 3.4.1 and section 3.4.2, we added some comparisons to other relevant studies conducted in hilly or lowland watersheds around the world.*

Other specific comments:

1) L19, L21, and other lines in abstract, too many details, I suggest deleting and shortening appropriately.

*Re: Thank you. We deleted some details and shortened the abstract properly.*

2) Too much general information was given on the first paragraph in the Introduction, which could be compressed. Some of the lines are confusing. For example, e.g., L41, why land use patterns (not changes) can alter surface roughness?

*Re: Thank you for pointing this out. We changed the "land use patterns" to "land use changes". In the first paragraph of section 1 Introduction, we also deleted some general information and instead added a few specific statements to improve this part.*

3) L98-101, an awkward sentence, please rehearse.

Re: *We rephrased and improved it (L113-115 in revised manuscript).*

4) L106, the first objective was coming suddenly, maybe because the model was not well introduced in the above text.

*Re: We provided a paragraph (the third paragraph of section 1 Introduction) to introduce the application of the hydrological model and particularly the wide applicability of the SWAT model in the land use impact studies, more importantly, we highlighted the high suitability of using the enhanced SWAT (i.e., SWAT$_{3s}$) to model hydrology and water quality in lowland areas. Therefore, the first objective sentence is better introduced.*

5) L163, it is confusing given that the following lines stated that different land use data were used for calibrating.

*Re: As stated in original L163 only the land use map of 2019 was used for model setup, model calibration and validation of streamflow. As we replied to your general comment (3) above, after streamflow was well simulated, the land use maps 2010 and 2019 were subsequently used for calibrating and validating water quality variables, respectively. We reworked section 2.3.3 and included a table to describe these details.*

6) L181, I would mention that suspended sediment is different from sediments as noted in L107.

*Re: Thank you very much for this helpful comment. The daily sediment load (in original L107) was calculated based on daily concentration of total suspended sediment concentration (in original L181) and measured daily streamflow. We added relevant information in the last sentence in section 2.3.2 Model databases and setup.*

7) L196, why TP/TN and sediments were calibrated for different years?

*Re: Actually, we calibrated TP, TN, and sediment in the same years (2009-2011) as we stated in original L196. But specifically, we calibrated TP and TN loads during the period of 08/08/2009-10/08/2011, and calibrated sediment load during the period of 30/10/2009-07/08/2011, which differs slightly due to the availability of measurements of them.*

8) Table 2 could be moved to the supplementary material.

*Re: Yes, we moved Table 2 to the supplementary material. It is Table S1 now.*

9) L221, More details should be given on how the land use data were used?

*Re: Yes, we included more details in this regard. We provided a table about the description of calibration and validation including how the land use data were used. In section 2.3.4, we implemented three- land use-scenario simulations. Each simulation was run using a land use map from 1987, 2010, or 2019, respectively. Further details were added in the section 2.3.4.*

10) L239, it is confusing on "between 1987, 2010, and 2019", please clarify.

*Re: Here we wanted to state respective changes between 1987 and 2010, between 2010 and 2019, and between 198 and 2019. We followed another reviewer's advice and clarified it by providing a detailed description of the calculation of response variables in the section Text S2 and Text S3 in the supplementary materials*

11) L246, what's mean of "by world's criterion"?

*Re: "Wold criterion" is a common terminology in PLSR method proposed by Svante Wold (Wold et al., 2001). It is often referred as a criterion for assessing the variable importance for projection (VIP) statistic. "by word's criterion" means "according to word's criterion". We changed "by word's criterion" to "according to Word's assessment criterion" in L274-275 in revised manuscript (no track-change).*

12) L268, it is a weak statement. Have you ever evaluated the performance of the model on a monthly scale?

*Re: Thank you! In fact, we did not evaluate the performance of the model for the monthly data in this case, because there are only 24 (calibration) or 12 (validation) monthly water quality observations, which is not a reliable basis for model evaluation. However, it can be expected that the model performance is better on a monthly scale, as over- and underestimations tend to balance out. This statement is in agreement with our modeling experience and the literature. We provided relevant references (in L293 in revised manuscript (no track-change)) to support our statement.*

13) Fig.3, should use different colors to better show the curves of simulated and measured data.

*Re: We modified the color scheme for simulated and measured values of streamflow and water quality, respectively. Specifically, we used grey for measured values and blue for simulated values.*

14) L301, what is the meaning of " agricultural grasses"?

*Re: "agricultural grasses" indicates grasses grown on fields by farmers for feeding livestock.*

15) Fig. 6, should add the legend of the box plots.

*Re: We added a legend to help explain the box plots.*

16) Fig. 7, maybe better to show the data in percent of change?

*Re: We chose to show absolute changes, as these absolute changes are used in the PLSR approach. We believe that it is better to keep the response variables (changes of water quality and quantity) and explanatory variables (changes of land use indicators) consistently in absolute values. Moreover, the absolute change allows the reader to better compare changes for different variables (e.g. if evapotranspiration is increased by a certain absolute value, water yield is probably decreased by the same absolute value). We would therefore prefer to keep showing absolute changes.*

17) L376, "rural landscape patterns" is confusing? Why does this matter?

*Re: Thank you for spotting this. The term may be confusing. We changed it to "changes of the rural landscape" in the current line L402 in revised manuscript (no track-change). As about 75% variations in sediment, TP, TN are explained by the changes of indicators mostly related to arable land, we can therefore state that the change of rural land use plays a key role in influencing water quality variations.*

18) L394, should denote the means of "AIa", "CONTIGAWa"….

*Re: Yes, we denoted the meaning of "AIa", "CONTIGAWa" right before them, respectively, they are shown in current line L425-L426 in revised manuscript (no track-change).*

19) L448-449, this sentence is not well fit with the paragraph, as the above lines are mainly related to management.

*Re: We agree with the reviewer, and removed this sentence.*

20) L473 and following lines, I don't think it is necessary to highlight the methodology, as the methods adopted here were not novel.

*Re: We agree with the reviewer, and removed this sentence to avoid highlighting the methodology here.*

**Third reviewer**

The manuscript subject is important for providing information on land use change and corresponding impacts to water quantity and quality. The authors carry out a modelling study and 3 years of field work in the Stör catchment in Germany. However, as is often with catchment studies, detailed spatial measured observations are missing over a longer time period to make specific causal links between the type and time of land use change and the corresponding hydrological impact. The authors use a mix of model simulations, water samples, and statistics (e.g. PLSR) to draw such links. Several points need to be addressed before the manuscript can be published because as it reads now, the reader is not able to determine if appropriate statistical methods were used to draw the conclusions made.

*Re: Thank you very much for reviewing this manuscript and providing constructive comments. We should highlight that the model accurately simulated daily long-term streamflow at the available three gauges in the catchment. We are therefore confident that the model is capable of representing spatio-temporal water fluxes in the catchment. Moreover, we agree with the reviewer that long-term measurements of water quality on a daily time step are rather scarce. However, for the study area observations for three time periods in three decades were available, which is a comparatively good data basis. We addressed the points raised by the reviewer to improve the manuscript and provided further details on the used methods.*

Introduction

The introduction can be tightened to highlight the scientific challenge and provide background information to frame the scientific question better. For example, especially the first part of the introduction until L 45 contains very general statements. The same applies to L 51-54, which can be re-written to be more to the point of the manuscript. See also L 72-73 that provides a very general statement, it would be more interesting to be more specific.

*Re: Thank you very much for the constructive comment. We revised and improved parts of the introduction to better highlight the scientific challenge and*

*framed the scientific question of this study. Particularly, we rewrote or improved the sentences as suggested by the reviewer. In response to another reviewer's comment, we also deleted some of these general statements in Introduction.*

The key information in the introduction is provided on L 86-89, yet it is not described in sufficient detail. What have other scientific studies found on this topic of using PLSR for land use change and hydrology?

*Re: Thank you for his very helpful suggestion. We described the key information on original L86-89 with more details, and meanwhile stated what previous scientific studies mainly found using PLSR for land use change and hydrology (added at the end of the fourth paragraph of section 1 Introduction).*

Overall, recent references are missing to show the state of the art.

*Re: We added some more recent references.*

Terminology

Several key terms used throughout the manuscript are not adequately defined. The term "landscape metric" first appears on L 63 and used several times throughout the paper, also in key places in the results and discussion, but it is never defined. Please provide a description and examples of what is meant exactly by "landscape metric".

*Re: This is indeed a key term and we provided a description of "landscape metric" (in current L41) and examples (in current L42-45) of explaining its meaning where it first appears in the revised manuscript (no track-change).*

Similarly, the term "patch" is used and not defined (is this a similar land use area?). What exactly is mean by a patch? In Table 1, the metric landscape patch index (LPI) is described using the word patch, which is also not helpful if "patch" is not described beforehand.

*Re: Thank you very much. Yes, "patch" denotes the homogenous areas of the landscape. We added the definition of the term "patch" where it appears at the first time (current L43) in the revised manuscript (no track-change).*

Some terms are not consistently used, for example on L 307 "PPLAND" is used and on L 401 "PLANDp" is used. Is there a difference in these terms? Would be useful to briefly describe the nomenclature used.

*Re: Thank you for spotting this. This is a typo. We took care of this, checked the rest of the manuscript and revised it carefully to make sure the terms are consistently used.*

PLSR

Section 2.4 is a key part of the study, yet it is poorly described. From L 236-248 the methodology is not explained in sufficient detail, below are just some examples:

*Re: Thank you very much. We worked on this section and added a few statements to introduce the method. Specifically, we clearly explained more about how land use changes are employed as predictors and how modeled dynamics in water quality and quantity are incorporated as response variables for PLSR. We put these detailed explanations in the Text S3 in the supplementary materials. In addition, we provided some necessary details on the cross-validation method of PLSR in the supplementary materials.*

The change in landuse was from 3 periods, yet only 3 years (1987, 2010, and 2019) are provided. This is confusing and must be changed in the manuscript to specify that time intervals were examined. Re-write to define the time steps.

*Re: Thank you. Actually, it is correct that we had only three available land use maps (1987, 2010, and 2019). We analyzed three time intervals, by analyzing the change in land use from 1987 to 2010, the change in land use from 2010 to 2019, and the land use change from 1987 to 2019. To clarify this, we explained it in more detail in section 2.3.4 (Model application). Moreover, the newly added details about predictor and response variables in the PLSR approach (Text S3) as well as calculation of land use changes (Text S2) in the supplementary materials should help to better understand how the land use changes are incorporated in this study.*

L 240-241 is difficult for the reader to follow: can an example be provided of the 50 random repetitions on 10 equal segments of the data set taken?

*Re: Yes, we provided an example to explain the method of 50 random repetitions on 10 equal segments in Text S3 in supplementary materials: For example, a dataset of 60 observations will be randomly split into 10 equal segments (one segment has 6 observations) for 50 times. For each of these 50 random repetitions the model will be run for each segment and return a performance score. The mean model performance score is calculated as the mean of the performance scores for the 10 segments and 50 repetitions, i.e., averaging 500 performance scores. This value is used as the final performance of cross-validation of PLSR.*

L 246-7 rephrase the sentence "By Wold's criterion…"

*Re: Yes, the sentence was rephrased to "According to Wold's assessment criterion, the predictor …" in L274-275 in revised manuscript (no track-change).*

L 64-65 define the difference between a landscape configuration metric and a composition metric.

*Re: Thank you. We defined this difference properly now. Landscape configuration metric is a quantitative indicator that describes specific spatial characteristics (i.e., spatial arrangement and distribution) of land patches, e.g., the connectedness degree or shape complexity of the land patches of one class. A composition metric normally refers to the number or area of land patches belonging to one class within the landscape, without considering the spatial characteristics of patches or their distribution in the landscape, e.g., areal percent of a land use class. We briefly included this difference in L41-45 in revised manuscript (no track-change).*

L 205 provide a brief description of the methodology by Guse et al (2020) cited

*Re: We provided a brief description of this method right after where it appears, i.e., in L214-215 in revised manuscript (no track-change).*

L 413 rephrase the sentence "Clumped and connected …"

*Re: Yes, we rephrased and improved the sentence, now it is "Agriculture in more clumped and connected land patches with fewer edges …" in L445 in revised manuscript (no track-change).*

If possible, a sketch would be helpful to visualize what is meant by the terms : LPI, AWISI, CONTIGAW, AI, IJI

*Re: Thank you very much. We agree that this will help and we provided a sketch as part of the supplementary materials in Figure S1.*

**Response to editor's comments**

E1: please use the terms consistently throughout your manuscript. For example, 'hydrologic' and 'hydrological', 'runoff', 'flow' and 'streamflow'.

*Re: We checked this and consistently use hydrological – unless it is part of 'hydrologic response units' which is a technical term. We use 'streamflow', 'runoff' only when referred to 'surface runoff', and 'flow' with regard to 'low flows', 'peak flows', and 'groundwater flow'.*

E2: please explain how SWAT3s differs from the standard SWAT in dealing with the German lowland catchment as you state in Section 2.3.1. What low flows and groundwater dynamics are better represented in SWAT3s?

*Re: at the time of development of SWAT3S, the standard SWAT did not allow for more than two groundwater aquifers, resulting in a limitation to depict fast and slow groundwater response. To this end, SWAT3S was developed. By subdividing the shallow groundwater aquifer into a fast and a slow shallow aquifer, it was capable to better represent groundwater contribution to streamflow in lowland catchments of Northern Germany. Although the latest versions of SWAT do allow for setting up more than two groundwater aquifers, we chose to use SWAT3S as it was specifically developed in the lowlands of Northern Germany.*

*We adjusted the text in section 2.3.1 as follows: "In comparison to the standard SWAT model application that uses two aquifers SWAT3S employs three aquifers by subdividing the original shallow aquifer from SWAT into a fast and a slow aquifer. SWAT3S was developed in the German lowland catchment of the Kielstau, where it better represented low flow periods of streamflow and groundwater storage and flow dynamics when compared to the original SWAT version (Pfannerstill et al., 2014). It was also successfully applied to the lowland catchment of the Treene proving its usefulness for modeling nutrients as well (Haas et al., 2017; Haas et al., 2016)."*

E3: WILL in 'Table 2' should change to 'WIL'

*Re: Thank you very much! Change applied to Table 2 and (former) Table 3.*

E4: Please revise the tense used in the text. You can use past tense since the work was done already. For example, L203: the parameters most likely to affect hydrological and water quality processes have been selected and their preliminary ranges have been defined. 'have been selected' should change to 'were selected'.

*Re: Thank you very much. Following the suggestion from one reviewer, we removed part of the description of model calibration and validation, instead, we provided these information in a table. We checked the tense used in the text and revised it where necessary.*

E5: The Section '2.3.3 Model calibration and validation' requires careful revision as it does not explain the methods clearly.

*Re: We restructured the entire section to provide more clarity: 1) We explain the stepwise procedure, 2) the used parameter ranges, 3) parameter sampling and model evaluation, 4) streamflow calibration, 5) calibration of sediment, total phosphorous, and total nitrogen. In response to reviewer's comment, we organized the description details of calibration and validation using the current Table 2. In the following, we provide detailed replies to your comments.*

For example, it is unclear how you calibrated streamflow using a fifteen-year time period from 1990 to 1991 and from 2007 to 2019, because you later stated "For each parameter set a model run has been performed, allowing for a warm-up period of 4 years."

*Re: For streamflow, the model was actually run from 1986-2019 (4 years warm-up + 15 years calibration + 15 years validation). This information is now available in Table 2. Our calibration procedure allows for using non-continuous periods for model evaluation. We also stated that "Calibration and validation periods (Table 2) were defined based on an equal representation of dry, normal, and wet years according to the annual precipitation." in L232-233 in revised manuscript (no track-change).*

You repeatedly stated 'lowland catchments but did not explain what is special about them and the relevance in using the SWAT model for lowland catchments.

*Re: Thank you for the comments! The special characteristics of lowland catchments are illustrated in the current L67-68, L76-77 in the newly added paragraph (i.e., the third paragraph) of "Introduction". Besides, the following information is included in this paragraph to highlight the suitability of using SWAT in this study.*

*"In lowland areas, the transport of water and nutrients is strongly influenced by flat topography and shallow groundwater tables in addition to the spatially heterogonous land use. The hydrological model SWAT has proven its suitability to model eco-hydrological consequences of spatio-temporal land use changes in lowland catchments (Guse et al., 2014; Pott and Fohrer, 2017b). Particularly in several lowland catchments in northern Germany, SWAT was extensively tested in impact studies. E.g., Lam et al. (2012) modeled the long-term observations of daily streamflow and nitrate load in the Kielstau catchment and found that diffuse source pollution (dominated by agriculture) contributed dominantly (95%) to nitrate load; In the Upper Stör catchment, Song et al. (2015) coupled SWAT with HEC-RAS to analyze temporal dynamics of sediment loads in subbasins covered by heterogonous land use conditions."*

*Moreover, we explained the importance of using SWAT3s for dealing with special hydrological characteristics of lowland catchments:*

*"Despite a high feasibility of SWAT modelling water quantity and quality, previous studies illustrated that the original SWAT version sometimes performed relatively poor for recession limbs and low flow periods of streamflow (Guse et al., 2014; Pfannerstill et al., 2014). In lowland catchments, groundwater contributes significantly to low flows and thus becomes a dominant component of streamflow (Pott and Fohrer, 2017b). To more accurately model low flows, an enhanced version of SWAT, SWAT3S, was developed in the Kielstau catchment, by conceptually separating the shallow groundwater aquifer of the original SWAT into a fast and slow shallow aquifer (Pfannerstill et al., 2014). SWAT3S was successfully used for modelling daily streamflow and nitrogen loads in a few German lowland catchments (e.g., Kielstau and Treene) by improving the representation of low flow periods (Haas et al., 2017; Pfannerstill et al., 2014). Given the aforementioned strength, SWAT3S is suitable for assessing the impacts of land use changes on water resources in lowland areas dominated by groundwater recharge."*

You used the word 'step-wise' calibration but did not explain what it means. Did you calibrate streamflow and water quality variables separately or in a certain step-wise order?

*Re: This is now clarified at the beginning of section 2.3.3, as follows: "The variables daily streamflow (1), sediment (2), TP (3), and TN (4) data were calibrated separately and stepwise. The number in the parentheses denotes their respective calibration order, i.e., streamflow was calibrated first, followed by sediment, TP, and TN."*

Was the parameter set selected for water quality variables based on parameter sets selected based on steamflows?

*Re: Thank you for the comment! The parameter sets were independent. But the model runs for water quality variables use the best set of streamflow calibration parameters. This should now become clear in the following sentences in the L234-238 in revised manuscript (no track-change): "Second, with the derived set of best streamflow parameters, model runs for 5000 different sediment calibration parameter sets were carried out and the best model run was selected based on the highest NSE. Third, this model was run for 5000 different sets of TP*

*calibration parameters and the best model run was similarly selected using the NSE. Finally, based on the so far derived parameters, another 5000 model runs for TN calibration were carried out and the best model run indicated by the highest NSE was selected."*

You stated "The calibration has been performed based on 8000 (stream flow) and 5000 (sediment, TP, and TN loads) parameter sets generated using Latin Hypercube Sampling method (Soetaert and Petzoldt, 2010)." Are the 8000 sets independent from the 5000 sets?

*Re: Yes, they are independent. To be clearer, we added the following statement in L217-219 in revised manuscript (no track-change): "For each of these 8000 (streamflow) and 5000 (sediment, TP, and TN loads) independent parameter sets, model runs were conducted each involving a warm-up period (four years), and evaluated using multiple performance criteria to select the best parameter set."*

Why did you use KGE to select the best set of parameters for streamflow and NSE for the remaining variables (L215). This needs to be explained.

*Re: NSE is sensitive to large values, and in cases of a high variability, it could lead to larger volume balance errors (Gupta et al. 2009). We chose the KGE for streamflow calibration to derive a good overall performance (KGE>0.75). Among these good model runs, we selected one that represented the periods of water quality measurements best (using a combination of RSE and KGE), following the rationale that the performance of the model for streamflow is particularly important in these periods, as it affects water quality predictions through the calculation of loads. This is described in the fourth paragraph in section 2.3.3 as follows:*

 *"For each of the three streamflow gauges, we pre-selected the parameter sets that yielded a KGE >0.75 for the streamflow calibration period. To accurately represent streamflow dynamics during the periods of water quality measurements (Aug. 2009 - Aug. 2011 and Oct. 2018 - Nov. 2019), the mean RSR for the five flow duration curve (FDC) segments during these periods was assessed and the best 300 streamflow parameter sets indicated by a low RSR were selected. From these 300 sets, the final parameter set yielding the highest KGE in these periods was selected."*

*When calibrating the model for water quality, we found that middle and low values of each water quality variable were generally well represented by the model, but we observed that the peak values of water quality variables were often underestimated in the calibration period, which had a significant effect on the overall performance. We therefore decided to use the NSE as it is more sensitive to large values and added the following sentence in this part in L238-239 to explain the selection more clearly:*

*"To accurately represent peak loads and their dynamics, the NSE was selected as single criterion for the water quality variables."*

In Line 163 you stated "The land use map for 2019 is used to build the model." but in the following Section 2.3.3 you stated three different landuse maps used for different time periods. This is confusing.

*Re: Yes, the model was set up using the land use map of 2019. And the model was calibrated and validated for streamflow using the land use map in 2019. It was calibrated for water quality variables using land use map 2010 and validated for them using the land use map of 2019. All of this information is now provided in Table 2. After the model was successfully calibrated for streamflow and water quality variables, we implemented scenario model runs for different land use maps in 1987, 2010, or 2019, respectively.*